# An ancient genome of *Streptococcus pyogenes* from a pre-Columbian Bolivian mummy

Guido Valverde [1,8] ✉, Mohamed S. Sarhan [1,2,3,8], Ryan Cook[4],
Omar Rota-Stabelli [5], Evelien M. Adriaenssens [4], Albert Zink [1,6,7] &
Frank Maixner [1] ✉

*Streptococcus pyogenes*, or Group A *Streptococcus* (GAS), is a human pathogen responsible for a range of diseases, from mild infections to severe illnesses. Despite its significance in modern clinical settings, little is known about the pathogen's evolutionary history or its presence in ancient human populations. Here, we present genomic evidence of *S. pyogenes* in the pre-Columbian Americas. We analysed a tooth from a naturally mummified individual dating to the Late Intermediate Period (1283–1383 cal AD), housed in the National Museum of Archeology (MUNARQ) in La Paz, Bolivia. Mitochondrial DNA analysis confirmed the host's Native American ancestry. Shotgun metagenomic sequencing and *de-novo* assembly enabled the near-complete reconstruction of an ancient *S. pyogenes* genome displaying close similarity to contemporary strains linked to pharyngitis. The genome contains core virulence genes, but prophages lack streptococcal pyrogenic exotoxins. Phylogenetic analyses place the strain at the base of modern *S. pyogenes* diversity, and Bayesian analyses indicate that most extant lineages diversified globally within the past ~5,500 years. Our results push back the confirmed presence of *S. pyogenes* in the Americas by several centuries and suggest that the pathogen circulated among Indigenous populations prior to the European contact.

*Streptococcus pyogenes*, also known as Group A Streptococcus (GAS), is a globally relevant human-adapted pathogen responsible for a wide spectrum of diseases—from mild infections such as pharyngitis (strep throat) to life-threatening conditions including necrotizing fasciitis, toxic shock syndrome, and serious post-infectious complications like rheumatic fever and glomerulonephritis[1-3]. Despite its critical impact on public health, our understanding of its long-term evolutionary history, particularly its presence and virulence in ancient human populations, remains virtually non-existent.

This knowledge gap is especially striking given the pathogen´s current prominence. Recent decades have seen a resurgence in GAS-related illnesses, including scarlet fever outbreaks and rising rates of invasive infections[2,4]. Advances in genomic epidemiology have revealed the emergence of highly virulent lineages and underscored the importance of mobile genetic elements—particularly prophages—in driving GAS evolution and virulence[5-8]. Scarlet fever, once a leading cause of childhood mortality prior to the antibiotic era, is caused by toxigenic strains of *S. pyogenes*. The disease was mentioned in the

[1]Institute for Mummy Studies, Eurac Research, Bolzano, Italy. [2]Institute for Biomedicine, Eurac Research, Bolzano, Italy. [3]Department CIBIO, University of Trento, Trento, Italy. [4]Quadram Institute Bioscience, Norwich Research Park, Norwich, UK. [5]Center C3A, University of Trento, San Michele all'Adige, Trento, Italy. [6]State Collection for Anthropology, SNSB, Munich, Germany. [7]Anthropology, Faculty of Biology, Ludwig-Maximilians-Universität (LMU), Munich, Germany. [8]These authors contributed equally: Guido Valverde, Mohamed S. Sarhan. ✉e-mail: guido.valverde@gmail.com; frank.maixner@eurac.edu

London Bills of Mortality (1657–1758)[9], together with other fevers, as the second most common cause of death at that time[10]. Historically feared for its rapid transmission and high fatality rates, scarlet fever has reemerged in recent years. Since late 2022, several European countries have reported significant outbreaks[11–13], predominantly afflicting children and often resulting in higher-than-expected mortality rates. This resurgence raises urgent questions about the continued threat posed by this pathogen despite medical advances. The pathogenicity of scarlet fever is driven in part by potent exotoxins (superantigens) that bypass normal immune regulation and provoke excessive immune responses, manifesting in symptoms such as rash, fever, and sore throat[14]. Not all *S. pyogenes* infections progress to scarlet fever, indicating that disease manifestation depends on a combination of host immune factors and the presence of specific bacterial virulence genes.

The closest phylogenetic relative of *S. pyogenes* is *S. dysgalactiae* subsp. *equisimilis* (SDSE), sharing extensive genomic homology and many homologous virulence determinants[15]. In contrast to *S. pyogenes*, that belongs to Lancefield serogroup A, SDSE commonly expresses Lancefield group C or G antigens, but still causes a similar spectrum of human diseases ranging from pharyngitis to invasive soft tissue infections[16]. Comparative genomics indicates frequent horizontal gene transfer between these species, contributing to overlapping pathogenic potential[17]. Mobile genetic elements such as bacteriophages may play a central role in the pathogen's evolution, reshaping population structure and enhancing virulence by facilitating horizontal gene transfer[3,5]. Prophages in modern GAS strains carry modular genetic cassettes that recombine frequently, allowing for the dissemination of key virulence factors (e.g., pyrogenic exotoxins) across subpopulations[18].

These insights have been pivotal for modern surveillance and vaccine development. Yet, all our understanding of this pathogen is based exclusively on modern strains. Despite the pathogen's modern worldwide prevalence and clinical importance, no genomic data from ancient *S. pyogenes* strains have yet been reported, leaving it unclear whether this pathogen has accompanied humans for millennia or emerged more recently. Was it already present in the Americas before European contact, possibly arriving with early human migrations across Beringia? Or did it spread globally only after the onset of European colonialism in the 15th century?

In this study, we provide genomic evidence of *S. pyogenes* in the pre-Columbian Americas. We analysed a tooth sample from a naturally mummified individual dating to the Late Intermediate Period (1283–1383 cal AD), housed in the National Museum of Archeology–MUNARQ in La Paz, Bolivia. Through shotgun metagenomic sequencing, we reconstructed the host mitochondrial genome, confirming the Native American ancestry, and applied *de-novo* metagenomic assembly to recover microbial genomes without reference bias. This approach led to the near-complete reconstruction of an ancient *S. pyogenes* genome–marking the earliest confirmed occurrence of this pathogen in the Americas.

The ancient genome assembly allowed us to apply modern *S. pyogenes* typing methods, infer the presence of virulence genes and prophages, and calibrate the pathogen's phylogeny in time to finally place this strain within a broader phylogenetic and functional framework.

## Results

We analysed a tooth sample (ID: 2730) from a pre-Columbian Andean mummy held in the anthropological collection of the National Museum of Archeology–MUNARQ in La Paz, Bolivia. The sample was recovered from a partially mummified human head of a male individual (approx. 18–25 years-old) (Fig. 1a). Radiocarbon dating (C14) assigned the individual to the Late Intermediate Period, i.e., 1283–1383 cal AD (with 95.4% probability) (Figs. 1a and S1,

Supplementary Data 1). Stable isotope analysis had a measured C:N ratio of 3,2 showing well-preserved tooth collagen. Carbon $\delta^{13}C$ showed −8,6‰ suggesting high maize consumption (e.g., $C_4$-based diet, typical for agricultural populations), and Nitrogen $\delta^{15}N$ showed 9,0 ‰, suggesting a low trophic level and a limited intake of animal protein (e.g., little meat or marine food) (Supplementary Data 1). We subjected the sample to DNA extraction and shotgun metagenomic sequencing, which resulted in ~6.8 million merged reads ("Methods"). Analysis of the retrieved DNA showed that 1.72% was human endogenous DNA (Fig. 1a, Supplementary Data 2). Next, we analysed the uniparental markers, i.e., mitochondrial DNA (mtDNA) and Y-Chromosome haplogroups in order to confirm the geographic origin and DNA authenticity of the individual. We report a B2 mtDNA haplogroup that falls within the genetic diversity expected in pre-Columbian Americas[19]. Not enough sequencing reads impaired the Y-Chromosome assignment, although molecular sexing[20] successfully confirmed the anthropological sex estimation (male, XY). The overall deamination of C to T was ~3.7%, while of G to A was ~5.1% (Fig. 1b). The low deamination level can be explained by the sample origin, coming from high-altitude environments with dry conditions, which have been previously shown to be less prone to DNA degradation[21]. Further microbial profiling using MetaPhlAn 4 displayed unusual high abundances of different opportunistic human pathogenic bacterial species, i.e., *Clostridium tetani* (37.5%), *Morganella morganii* (29.5%), *Streptococcus pyogenes* (11.6%), *Clostridium botulinum* (6.7%), *Asaccharospora irregularis* (6.4%), and *Paeniclostridium sordellii* (4.9%) (Fig. 1c, Supplementary Data 3). This is in addition to other less abundant commensal oral species, e.g., *Desulfobulbus oralis* (0.08%) and *Actinomyces dentalis* (0.005%) (Fig. 1c).

The detection of *S. pyogenes* was particularly significant; despite its prevalence in modern outbreaks, this pathogen has not yet been detected in ancient times. Hence, we opted to devote particular focus to this species to understand its evolutionary trajectories. Therefore, we decided to perform a metagenomic screening on the Ancient Metagenomics Directory AMDir[22] using MetaPhlAn 4, which revealed the presence of *S. pyogenes/dysgalactiae* in human samples from different time periods (ranging from 4000 to 200 years BP) in Europe and Africa, as well as in gorilla samples from museum collections (Fig. 1d). However, in nearly all cases, the available genomic data from these samples were insufficient to reconstruct complete genomes and thus precluded further genomic analyses (Supplementary Data 4). Only a 200-year-old dental calculus sample from a gorilla (*Gorilla beringei beringei*) from Congo yielded sufficient data to reconstruct a high-quality *S. dysgalactiae* genome (completeness = 96.26%, contamination = 0.12%)[23] (highlighted in Supplementary Data 4).

We then performed *de-novo* metagenomic assembly on the Bolivian sample to reconstruct a reference-free genome of *S. pyogenes*. The *de-novo* assembly resulted in the reconstruction of 7 high-quality bacterial metagenome-assembled genomes (MAGs), including *C. tetani*, *M. morganii*, and *S. pyogenes* (Supplementary Data 5), in addition to other 3 medium-quality genomes, including *A. irregularis* (Supplementary Data 5). The completeness of the assembled *S. pyogenes* genome was 99.98%, with a 0.28% contamination as estimated by CheckM2 ("Methods"). The total size of the MAG was 1,703,496 bases distributed on 29 contigs ranging in size from 2,244 to 250,505 bases with a GC content of 38.44% (Table 1). Considering the minimum information about MAGs[24], the ancient, assembled genome is of high-quality, given that the completeness is >90% and contamination is <5%, and the presence of rRNA genes (23S, 16S, and 5S) and 24 tRNAs (Table 1). The overall coverage of the reconstructed genome is 8–16X (Fig. S2). Similar to human DNA, analysis of ancient DNA damage revealed low levels of deamination of C-to-T and G-to-A (Fig. S2).

To systematically and functionally assign the ancient *S. pyogenes* strain, we applied Multi-locus Sequence Typing (MLST), *emm*, and pilin typing schemes that are commonly used to classify modern

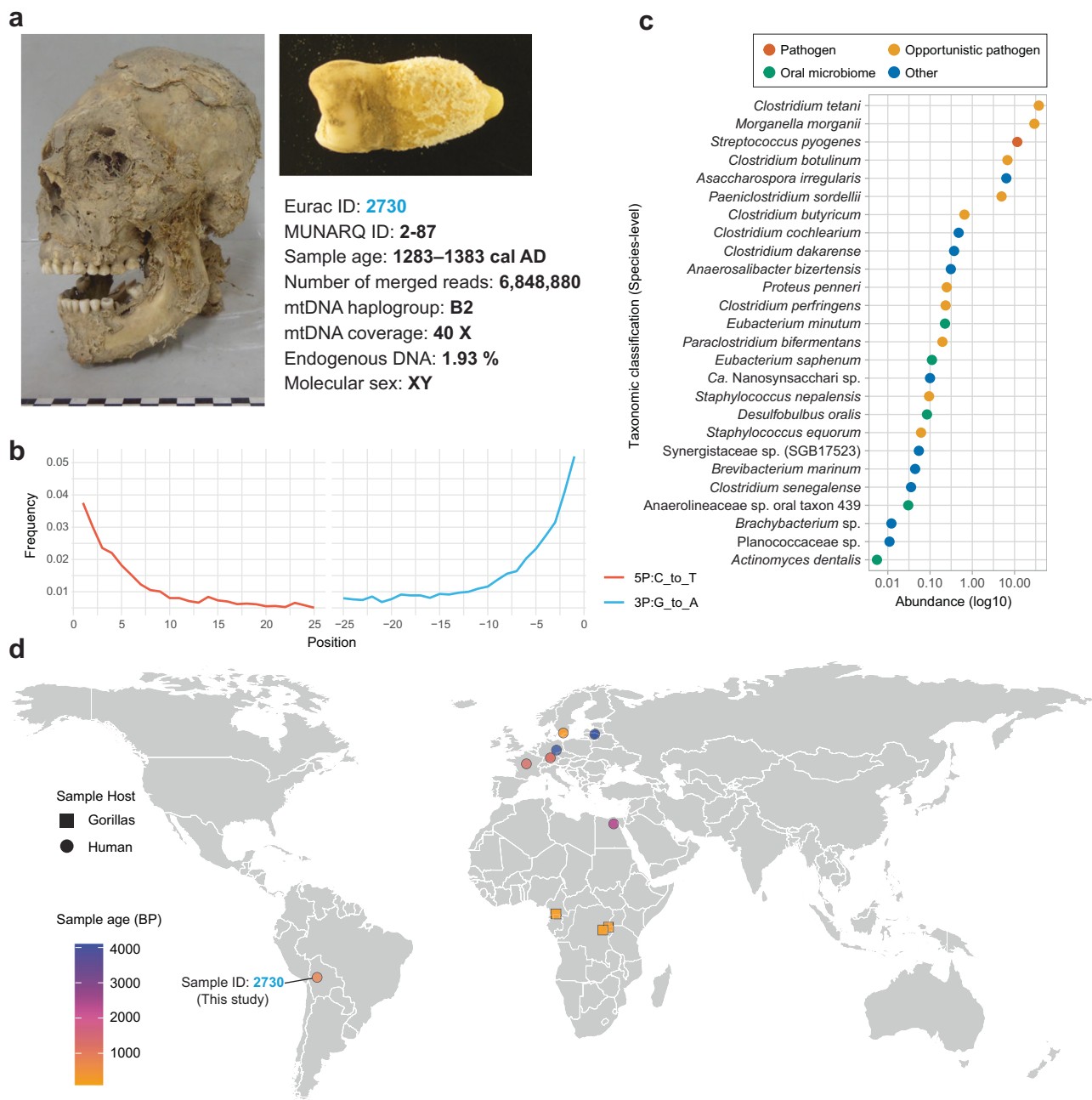

**Fig. 1 | Ancient Bolivian *S. pyogenes* case and comparative analysis. a** Dental sample from individual Eurac ID: 2730, a pre-Columbian mummy with mtDNA haplogroup B2. Endogenous DNA content showed 1.93%, and molecular sex was determined as XY. **b** DNA damage of human DNA from the sample, showing characteristic deamination pattern analysed using MapDamage 2. **c** Taxonomic classification of the microbial community in the sample based on MetaPhlAn 4, highlighting the presence of *S. pyogenes* alongside other pathogens, opportunistic pathogens, and oral microbiome members. **d** Screening of the Ancient Metagenome Directory (AMDir) using MetaPhlAn 4, reporting positive ancient *S. pyogenes/dysgalactiae* cases across different geographic regions and time periods, with sample ID: 2730 from this study marked in light blue font.

*S. pyogenes* strains[25,26]. MLST of seven housekeeping genes assigned the ancient strain to a distinct sequence profile, that has no direct modern equivalents and possibly represents an ancestral lineage of contemporary strains (Supplementary Data 6). The most similar modern *S. pyogenes* isolates come from remote aboriginal island communities in tropical Australia, sharing three out of seven MLST alleles with the ancient strain[27]. Additional *S. pyogenes* typing schemes target the *emm* gene encoding the surface-exposed streptococcal M protein, and pilin adhesin and backbone genes[28]. Both M proteins and pili are key virulence factors and targets of host protective immunity[29].

The *emm* gene is located in the *mga* regulon with genes encoding additional M-like proteins such as *mrp* and *enn*, and *scpA* encoding a C5a peptidase[30]. In the assembled Bolivian genome, the *mga* regulon genes are distributed to two different contigs, contig 17 that carries the transcriptional activator gene *mga* and a partial *emm* gene, and contig 13 that encodes the *emm*-like gene *enn* and the C5a peptidase gene *scpA* (Fig. 2b). Considering the size range of 220–513 amino acids in modern M protein sequences[31], the *emm* gene of the ancient Bolivian strain is nearly complete (1161 bp, 387 aa in length) lacking only the 3′ gene part that encodes the D-repeat region of the M protein (Fig. S3).

**Table 1 | Assembly statistics comparing individual (metaSPAdes, MEGAHIT) and merged (Flye) assemblies of the ancient *S. pyogenes* genome**

| | | MEGAHIT | metaSPAdes | Flye |
|---|---|---|---|---|
| Assembly quality | Completeness (%) | 99.98 | 99.99 | 99.98 |
| | Contamination (%) | 0.28 | 0.29 | 0.28 |
| Assembly statistics | Total length (bp) | 1707330 | 1695810 | 1703496 |
| | Total length (> = 25,000 bp) | 1585200 | 1483076 | 1620257 |
| | Total length (> = 50,000 bp) | 1263868 | 1163145 | 1336590 |
| | Number of contigs | 37 | 38 | 29 |
| | Largest contig (bp) | 250506 | 208929 | 250505 |
| | N50 (L50) | 70736 (8) | 85333 (8) | 99718 (7) |
| | N90 (L90) | 26863 (24) | 21758 (23) | 29130 (19) |
| Gene content | Coding Density | 0.869 | 0.873 | 0.87 |
| | GC Content | 0.384 | 0.384 | 0.384 |
| | Total Coding Sequences | 1637 | 1631 | 1630 |
| | CRISPR | 1- with 4 spacers | 1- with 12 spacers | 1- with 12 spacers |
| | tRNA | 38 | 22 | 24 |
| | rRNA | 3 (16S, 23S, and 5S) | 0 | 3 (16S, 23S, and 5S) |

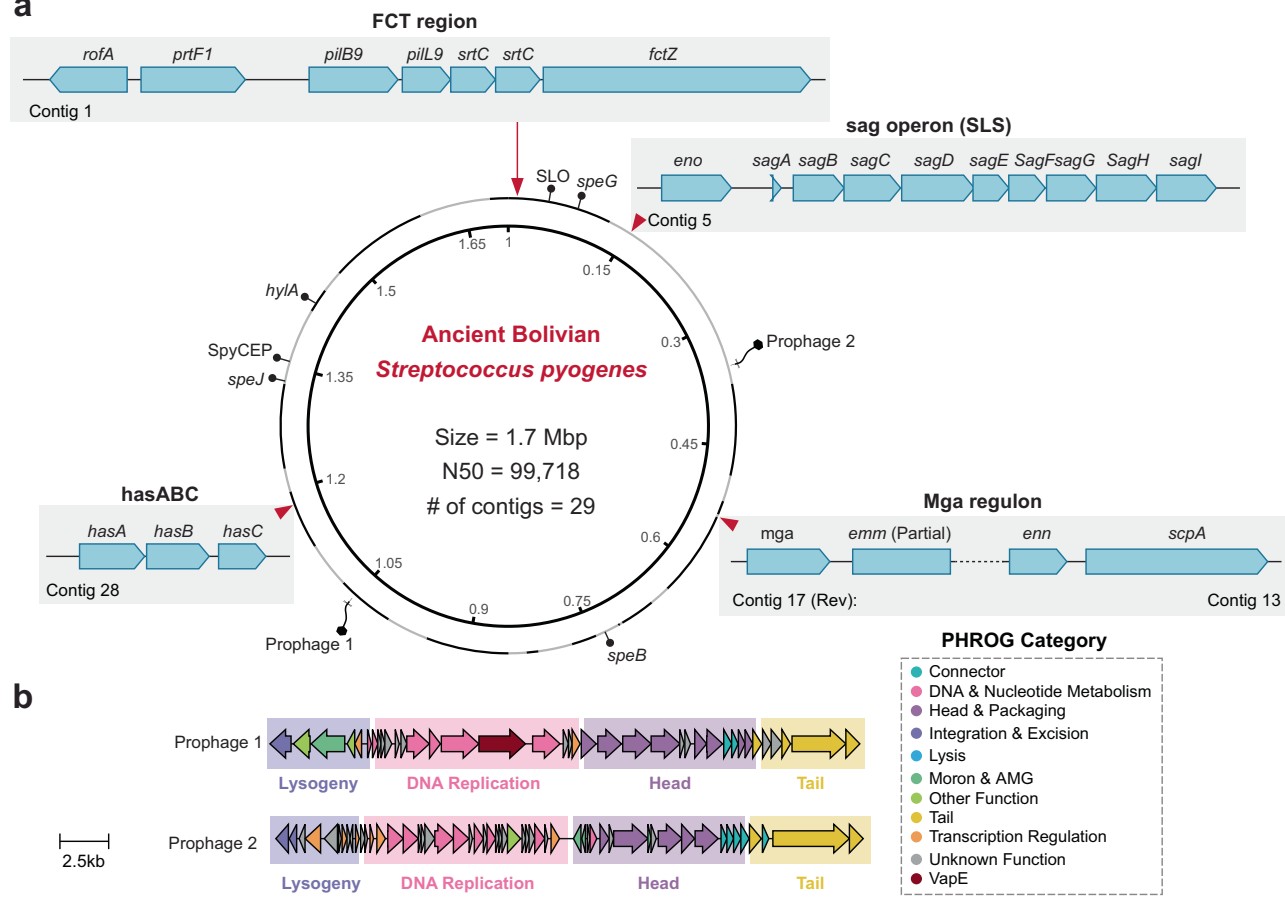

**Fig. 2 | Genomic map and key genetic regions of the ancient Bolivian *S.pyogenes* strain. a** Circular representation of the ancient Bolivian *S. pyogenes* genome assembly. The genome consists of 29 contigs with a total size of 1.7 Mbp and an N50 of 99,718 bp. Contigs are indicated as alternating black/gray colors in the outer circle. Key genetic regions and virulence factor loci are highlighted with expanded views showing their genomic context (For details, please refer to Supplementary Data 10): the FCT (fibronectin-collagen-T antigen); the sag operon (streptolysin S locus, SLS); the hasABC hyaluronic acid capsule biosynthesis operon; and the Mga regulon. Prophage insertion sites are marked with phage symbols. **b** Structural organization of the two complete prophages identified in the ancient Bolivian *S. pyogenes* genome. Prophage 1 (top) and Prophage 2 (bottom) are shown with their functional modules color-coded according to the PHROG (Prokaryotic Virus Remote Homologous Groups) classification system. Comparisons against the closest phage genomes from the EnteroBase database are shown in Fig. S4.

This gene part could not be assembled with the currently available data and assembly strategy. Nevertheless, *emm* typing targeting 5′ regions of the gene was possible and assigned the ancient *S. pyogenes* strain to the modern *emm* type 46 (79% nucleotide identity) (see Fig. S4, Supplementary Data 7). However, based on the currently existing typing criteria, the ancient strain should be assigned to a unique *emm* type, having less than 92% nucleotide identity over the first 90 bases encoding the mature M protein to any other *emm* type[32]. Next, we extended our strain typing to the *emm* pattern analysis that groups *emm* types into functional patterns (A–C, D, E) that correlate with tissue tropism, ecological niche, and pathogenic behavior[25]. Based on the chromosomal arrangements of *emm* and *emm*-like genes (*mga-emm-enn-scpA*, Fig. 2b) and their subfamily (SF) forms (*enn* SF-1), the ancient Bolivian strain can be associated with the *emm* pattern group A-C, being most likely a *S. pyogenes emm* pattern B strain. The analysis of a global collection of Group A *Streptococcus* revealed that the majority of modern strains belong to *emm* pattern groups D and E, whereas *emm* pattern B or C strains are nowadays only rarely observed[31] (Supplementary Data 8). Interestingly, the modern strains displaying the highest sequence identity to the partial Bolivian *emm* gene and that share a similar configuration of the *mga* regulon mostly come from the Americas or Oceania (Supplementary Data 8 and S9, Fig. S3).

Further analysis of the fibronectin-binding, collagen-binding, T antigen (FCT) region revealed that the ancient Bolivian *S. pyogenes* strain harbors a complete Rof regulon on contig 1, including the transcriptional regulator *rofA*, the fibronectin-binding protein *prtF1*, the backbone pilin *pilB9* (allele 15), the linker pilin *pilL9*, two pilus biosynthesis enzymes (sortases), and a putative LPxTG-anchored surface protein (*fctZ*) (Fig. 2b). Based on the chromosomal organization of the FCT genes, the absence of an adhesin pilin gene (pilA), and the presence of pilB9, this strain is classified as carrying an FCT-9 region typical of group A streptococci (GAS)[28].

In addition to the surface-associated M- and T-proteins, the ancient *S. pyogenes* strain holds a diverse array of virulence genes that may enable it to colonize, invade, and evade the host immune system (Fig. 2, Supplementary Data 10, S11)[33]. The hyaluronic acid capsule (hasABC operon) encoded on contig 28 mimics host tissue and contributes to immune evasion. The full set of genes (*gacA-L*) of the group A carbohydrate (GAC) gene cluster is present on contig 5 with high homology, i.e., >99% proteins identity, to modern GAS strains (Supplementary Data 10). Among the virulence-associated enzymes, the ancient strain carries the cell-wall-anchored proteases C5a peptidase (ScpA, contig 13) and chemokine protease (SpyCEP, contig 20), and the secreted proteases hyaluronidase (HylA, contig 4) and pyrogenic exotoxin B (SpeB, contig 19). In addition, the ancient strain encodes the two important *S. pyogenes* exotoxins streptolysin O (SLO, contig 1) and streptolysin S (SLS, contig 5), that both promote host cell lysis. Interestingly, the ancient strain carries only two streptococcal super-antigens, the streptococcal pyrogenic exotoxin (Spe) type G (SpeG, contig 1) and J (SpeJ, contig 20), that are known to be encoded on the core chromosome of modern *S. pyogenes*[34,35]. Other important streptococcal superantigens, such as SpeA, SpeC, or the streptococcal superantigen SSA, which are all encoded in contemporary strains on bacteriophage elements, are missing in the ancient *S. pyogenes* strain.

There were two high-quality prophages identified in the ancient *S. pyogenes* genome (Fig. 2, Fig. S4). Despite the absence of super-antigens encoded on these two prophages, one prophage harbored a virulence determinant, Virulence-Associated Protein E (VapE), that was previously associated with virulence in a mouse model of *S. pneumoniae* infection[36]. Comparative analyses revealed that both prophages are closely related to previously characterized prophages from a large-scale study of *Streptococcus*[28,36]; however, the prophages in the ancient strain appear to lack a specific region that was present in those near relatives (Fig. S4). Follow-up searches recovered four additional

prophage contigs corresponding to these missing segments, but it remained ambiguous which two belonged to which prophage, because each prophage shared highly conserved regions that caused assembly breaks. Inspection of the assembly graph confirmed multiple valid pathways for contig arrangement. Despite this uncertainty, both prophages likely represent complete genomes with all necessary machinery for excision and replication. Further screening of publicly available *S. pyogenes* genomes indicated that these two prophages, or close variants of them, are widely distributed across the modern *S. pyogenes* phylogeny. This pattern of presence or absence across distinct bacterial clades highlights the role that prophages play in shaping the accessory genome and potentially driving the evolutionary trajectory of *S. pyogenes*. Comparative genomic analyses of prophage architecture (Fig. S4) uncovered a conserved modular organization—lysogeny, DNA replication, head, tail, and lysis genes—punctuated by mosaicism and recombination events. Interestingly, in some related strains, the DNA replication region contained a larger number of shorter predicted CDS, suggesting partial degradation that could lead to non-infective or "cryptic" prophage states. The detection of structural rearrangements, evidence of circularization, and functional gene conservation in these prophages collectively underscores their historical and ongoing relevance to *S. pyogenes* evolution and virulence potential.

Additional comparative analysis identified two antibiotic resistance genes in the ancient *S. pyogenes* genome using ABRicate v1.0.0 with the CARD database (Supplementary Data 12). The *lmrP* gene, encoding a multidrug MFS transporter, exhibited 99.9% identity and 97.5% coverage, and is associated with resistance to macrolides, lincosamides, streptogramins, and tetracyclines. Furthermore, the *mef(E)* gene, another macrolide efflux pump, was recovered with 95.9% identity and 82.4% coverage. Both genes were complete, with no evidence of frameshifts or premature stop codons, suggesting they were intact in the ancient strain. Their functional properties, however, remain to be determined.

The expression of the above-mentioned virulence genes, the modulation of antibiotic resistance and stress-response genes, and the control of prophage and other mobile genetic element genes is mediated by specific two-component regulatory systems (TCS)[37]. Group A *S. pyogenes* possesses up to 14 TCS to adapt the bacterial behavior to host and environmental conditions. The ancient Bolivian strain encodes for 11 TCS, including the classic complete *covR/S* TCS that acts as a global control for many virulence genes (Supplementary Data 13). The three TCS *salKR*, *silAB*, and srtRS that are missing in the ancient strain are described to be completely absent in other modern GAS genomes as well[37].

To further investigate the evolutionary placement of the ancient *S. pyogenes* strain, we performed a phylogenetic analysis with the HC254 clusters of modern *S. pyogenes*, *S. dysgalactiae,* and *S. canis* assemblies from the EnteroBase database (Supplementary Data 14). This analysis was based on the core genome (Please refer to Methods for details) of 605 conserved genes in >99% of the genomes at 95% similarity. The phylogenetic tree placed the pre-Columbian Bolivian strain basal to the modern *S. pyogenes* diversity (Fig. 3a). The second *Streptococcus* genome from the 200-year-old gorilla was assigned to the closest phylogenetic relative of *S. pyogenes*, *S. dysgalactiae* subsp. *equisimilis* (SDSE) (Fig. 3b). To further support the phylogenetic placement of both ancient strains, we aimed to identify species-specific KEGG modules using pangenome analysis ("Methods"). The final set of KEGG modules clearly indicate metabolic differences between the three species. Either complete KEGG modules or certain genes within modules are present in one species but not the other (Fig. 3b and Fig. S5, Supplementary Data 15, S16). As previously reported by Xie and colleagues[17] in their pangenome comparison of modern *S. pyogenes* and SDSE strains, we could confirm in our dataset the absence of the modules encoding

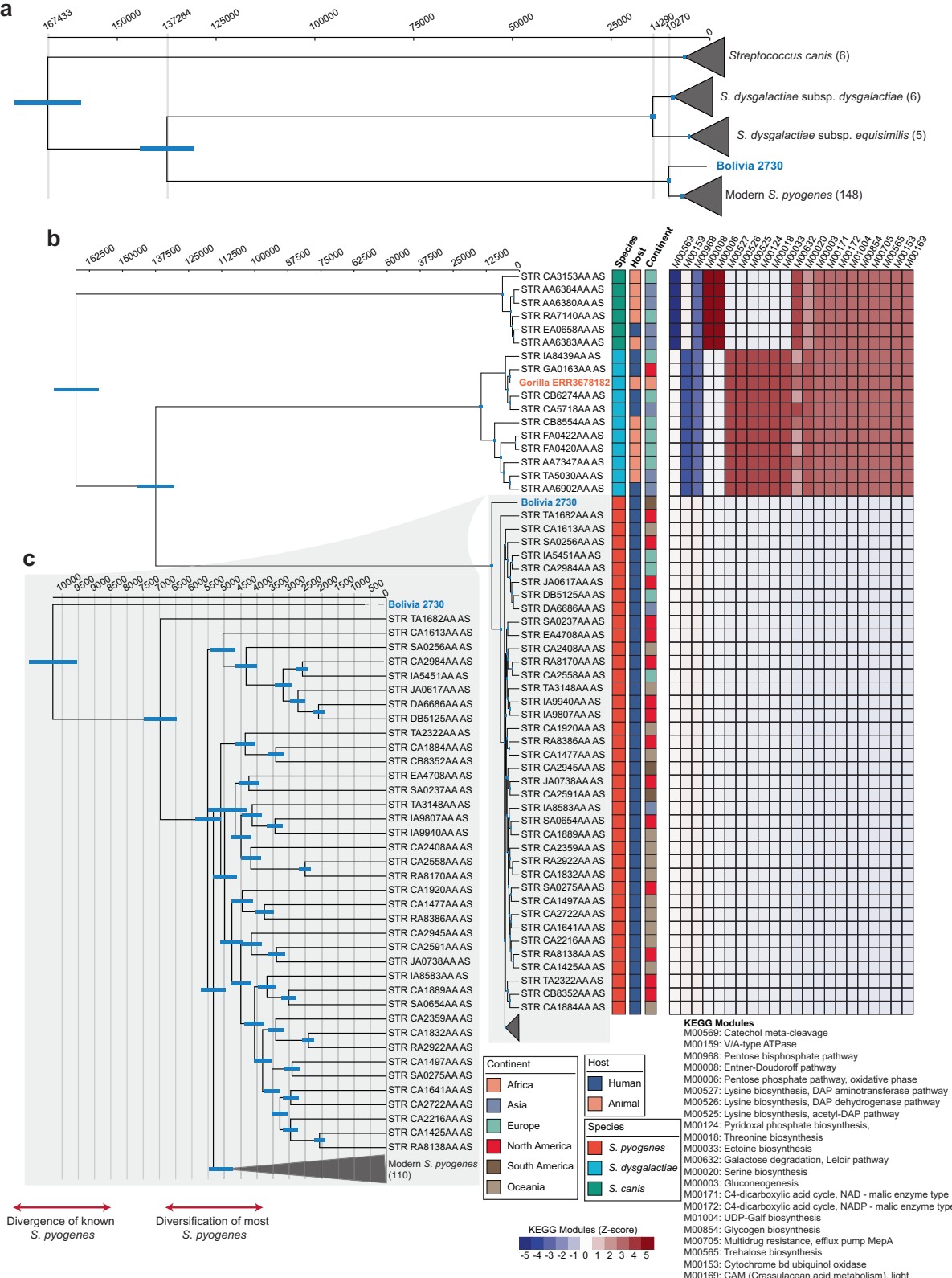

glycogen biosynthesis (M00854) and threonine biosynthesis (M00018) in both the ancient Bolivian and modern *S. pyogenes* strains. Furthermore, unique to the ancient and modern *S. pyogenes* strains were the presence of multiple genes encoding V/A-type ATPases (M00159). There are additional differences in the metabolic gene set between the three *Streptococcus* species, such as the unique presence of possible *cydAB* genes that encode subunits of

the Cytochrome bd ubiquinol oxidase in SDSE, *S. canis,* and the gorilla strains. However, we do not see any indications for genes that occur only in the ancient Bolivian or gorilla strain.

In summary, the phylogenetic placement of the ancient Bolivian strain, basal to the modern *S. pyogenes* and the gorilla strain into the SDSE diversity is highly supported on the pangenome level by the presence of species-specific genes in both ancient genomes.

**Fig. 3 | Phylogenomic analysis and functional characterization of *Streptococcus* species. a** Time-calibrated phylogeny of major *Streptococcus* lineages reconstructed using BEAST molecular clock analysis. The tree shows the evolutionary relationships between *S. canis* (*n* = 6), *S. dysgalactiae* subsp. *dysgalactiae* (*n* = 6), *S. dysgalactiae* subsp. *equisimilis* (*n* = 5), and modern *S. pyogenes* (*n* = 148). Blue bars on nodes indicate the 95% highest posterior density (HPD) intervals for divergence time estimates. The ancient Bolivian mummy strain Bolivia 2730 (highlighted in blue) represents a deep-branching lineage within the *S. pyogenes* clade. Time scale shown in years before present (top axis). **b** High-resolution time-calibrated phylogeny of *S. pyogenes* strains, including the ancient Bolivian mummy sample Bolivia 2730. The phylogeny demonstrates the placement of Bolivia 2730 as a basal lineage that diverged prior to the diversification of modern *S. pyogenes* strains. Heatmap (right) displays the counts of enzymes per each KEGG modules, that are significantly differential across samples ("Methods"). Samples are annotated by the continent of origin (Africa, Asia, Europe, North America, South America, Oceania), host type (Human, Animal), and species classification. Module abundance is shown as z-scores. Top 22 modules with significant differential abundance between species based on Kruskal–Wallis test and FDR values are shown (For full list, please refer to Supplementary Data 15, S16). Blue bars on nodes indicate 95% HPD intervals. Bolivia 2730 and Gorilla ERR3678182 samples are highlighted. **c** Detailed phylogeny of 110 modern *S. pyogenes* strains with expanded view of phylogenetic relationships. The tree reveals the genetic diversity within contemporary *S. pyogenes* populations and demonstrates the ancient position of Bolivia 2730 relative to extant strains.

Finally, to obtain improved estimates of the evolutionary history of *S. pyogenes* and to infer divergence times among its closest related species, we applied a Bayesian phylogenetic framework using BEAST. Time estimates of the full dataset under most fitting analytical parameters (see "Methods") returned an age for the split of *S. pyogenes* and *S. dysgalactiae* spp. at 137,264 years (95% HPD: 124,428–149,847 cal years BP) (Fig. 3a). The timing of the Most Recent Common Ancestor (MRCA) of all *S. pyogenes* strains (ancient Bolivian strain and modern known *S. pyogenes*) is estimated to be 10,270 years (95% HPD: 9314–11,218 cal years BP). The diversification of most modern *S. pyogenes* lineages took place in the past 5500 years of human history (Fig. 3C). Time estimates are robust to demographic model variation and outgroup exclusion but become more recent when subsampling the dataset (Fig. S6c). In all cases, the split between the ancient Bolivian strain and the remaining modern *S. pyogenes* post-date by more than 15,000 years the supposed migration of humans into the Americas via the Bering Strait at circa 22,000 years ago.

## Discussion

This study presents genomic evidence of *Streptococcus pyogenes* (Group A Streptococcus) in a pre-Columbian Andean mummy from Bolivia. Radiocarbon dating placed the individual within the Late Intermediate Period (1283–1383 cal AD), a time marked by increasing regional interaction and sociopolitical complexity in the Andean region[38]. The isotopic profile of individual ID:2730 (high $\delta^{13}C$, low $\delta^{15}N$) indicates a maize-dominant diet with low trophic level protein intake, consistent with a sedentary agricultural lifestyle. This subsistence mode is associated with increased population density and reduced mobility, environmental factors known to facilitate the transmission of potential pathogens (i.e., *S. pyogenes*). Analyses of stable isotopic data in the Andean region were used to infer nutritional stress, pathological conditions[39], and migration patterns that spread diseases linked to evidence of societal collapse affecting health status[40]. Limited dietary diversity (implied by high maize reliance) may have compromised nutritional status in the Andean region[41], which could in turn impact immune function and susceptibility to such ancient infections or potential outbreaks in the past.

Systematic functional characterization of the ancient Bolivian *S. pyogenes* strain using modern molecular typing schemes reveals close similarity to contemporary *emm*46 type strains belonging to *emm* pattern group A–C. Although *emm* pattern A–C strains are less common overall than pattern D and E strains, they account for a substantial proportion of pharyngitis cases (~47%) while representing only a minor fraction of impetigo isolates (~8%)[42,43]. This epidemiological distribution has led to their classification as "throat specialists" in contrast to pattern D strains, which are primarily associated with skin infections. Additionally, the ancient strain lacks the serum opacity factor (SOF)[26], a defining feature of SOF-negative *emm* pattern A–C lineages. SOF-negative strains show a strong association with pharyngitis rather than impetigo and display marked seasonal dynamics, with increased prevalence during cooler months. These patterns closely align with the cold, dry environmental conditions of the Bolivian highlands, which are favorable for S. *pyogenes* pharyngitis outbreaks and contrast with the climatic conditions typically favoring skin infections.

Further evidence for throat specialization is provided by analysis of the FCT region, which reveals an FCT-9 configuration. FCT-9 isolates lack pilin adhesin genes and have been associated with an increased propensity to cause pharyngeal infections in modern populations[28]. Taken together, the convergence of *emm* pattern assignment, SOF negativity, FCT region composition, and environmental context strongly indicates that the ancient Bolivian *S. pyogenes* strain was primarily adapted to cause pharyngitis. These findings suggest that ecological specialization toward throat infection was already established in *S. pyogenes* lineages circulating in ancient high-altitude Andean populations, underscoring the long-standing evolutionary stability of tissue tropism in this pathogen.

The identification of core virulence genes indicates that the ancient strain possessed a pathogenic repertoire broadly comparable to that of contemporary *S. pyogenes*, supporting its classification as a disease-causing bacterium. Key surface-associated structures, including M and T proteins, the hyaluronic acid capsule, as well as secreted and cell wall–anchored proteases and exotoxins[33], appear to have remained remarkably stable over time, suggesting long-term evolutionary conservation of essential mechanisms for host interaction and immune evasion. In contrast, the ancient strain lacks streptococcal pyrogenic exotoxins encoded by lysogenic bacteriophages, which are commonly present in modern lineages. These phage-associated superantigens are potent drivers of immune overstimulation and are strongly linked to scarlet fever, streptococcal toxic shock syndrome, and severe invasive disease[34].

*S. pyogenes* lineages can be grouped into related clusters ("phylogroups") defined by whole genome clustering methods[5,44]. These phylogroups are not genetically static, as occasional recombination occurs via mobile genetic elements such as prophages, plasmids, and transposons, contributing to variation in virulence, antibiotic resistance, and immune evasion. Our analysis of the ancient *S. pyogenes* prophages reveals mobile genetic elements dating back 700 years that retain a complete lytic gene repertoire, whereas some modern counterparts show genomic degradation. The prophage mosaic structure highlights the long-term role of horizontal gene transfer, with conservation across selected lineages despite geographic separation[45].

The detection of macrolide efflux pump genes *lmrP* and *mef(E)* in the ancient *S. pyogenes* genome provides compelling evidence for the long-standing presence of antibiotic resistance determinants prior to the modern clinical use of macrolides. These genes are commonly found in contemporary clinical isolates and are often associated with mobile genetic elements, suggesting that horizontal gene transfer may have played a role in their ancient dissemination. This finding supports the idea that resistance genes are ancient and naturally present in microbes, shaped by long-term ecological competition rather than just modern antibiotic use[46].

Phylogenetic analysis shows that the ancient Bolivian *S. pyogenes* strain occupies a basal position and lies outside the diversity of known modern lineages. Despite this deep divergence, the strain possesses many genetic features associated with virulence in contemporary *S. pyogenes*, indicating that key pathogenic traits were already established early in the species' evolutionary history. The Bolivian strain may therefore represent either an extinct lineage or a genomically unsampled clade that no longer circulates in present-day populations.

Molecular dating suggests that the split between this ancient Bolivian strain and all other sampled *S. pyogenes* lineages occurred approximately 10,000 years ago. Archeological evidence and excavations in the Bolivian highlands have revealed an extensive number of Archaic period occupations. While radiometric dates are currently limited, the available evidence indicates a substantial human presence in the region starting as early as roughly 13,000 cal BP[47]. As humans first moved into the Andes, they underwent a sophisticated process of adapting both their physical biology as well as their social customs[48]. Both these processes and the strategic evolutionary responses, like potential pathogen adaptations and interactions, were developed to challenge and thrive in such a unique landscape. The divergence between modern and ancient *S. pyogenes* strains post-dates the widely accepted migration of humans into the Americas via the Bering Strait approximately 22,000 years ago[49]. Genomic evidence shows multiple gene flow events and regional population turnover across the Americas during the Holocene (<10 kya)[50,51], suggesting dynamic intra-continental population structure that may have influenced the pathogen's spread. While this timing is compatible with scenarios involving zoonotic transmission from an as-yet unidentified host or the introduction of an ancestral strain during post-glacial migrations of Siberian populations into the Americas within the past 10,000 years[52], these possibilities remain speculative. The basal phylogenetic placement of the ancient Bolivian strain and reconstructions of the most recent common ancestor may be consistent with an American origin of the pathogen; however, these observations alone are insufficient to support such a conclusion. Basal positioning can also result from the extinction of lineages in other regions or from incomplete sampling.

Consequently, additional global representatives of ancient and modern *S. pyogenes* genomes will be required to more rigorously evaluate the robustness of the inferred divergence dates and phylogenetic relationships. By expanding our search for *S. pyogenes* to other publicly available ancient DNA datasets, we detected the pathogen in humans as early as 4000 years ago in Europe and Africa. In addition, we observed traces of SDSE DNA, the closest phylogenetic relative of *S. pyogenes*, in gorillas around 200 years ago, suggesting a complex evolutionary history of both pathogens. The presence of *S. pyogenes* across geographic regions and time periods raises the possibility that it was carried over by human populations during their migrations, contributing to its global distribution. Our current Bayesian phylogenetic analyses indicate that the majority of modern *S. pyogenes* lineages diversified much more recently, largely within the past ~5,500 years of human history. This recent diversification likely reflects major changes in human population structure during the late Holocene in different parts of the world[53]. Rapid population growth, increasing population density, and expanding social connectivity would have enhanced transmission opportunities and sustained large effective pathogen population sizes. These conditions promote the accumulation of mutations, increase opportunities for recombination, and support the long-term persistence of successful lineages. In addition, strong immune-mediated selection—particularly targeting surface antigens such as the M protein—likely drove frequency-dependent turnover, enabling the maintenance of high antigenic diversity over time[5]. The acquisition of mobile genetic elements, including prophage-encoded virulence factors such as superantigens, further facilitated episodic clonal expansions and lineage replacement. Together, these evolutionary processes help explain how modern *S. pyogenes* lineages can exhibit relatively shallow apparent ancestry despite the species having a much deeper evolutionary origin. Our results highlight the value of ancient genomics in tracing the long-term evolution and global spread of clinically important pathogens. The detection of this pathogen expands our understanding of infectious diseases in ancient populations and illustrates the feasibility of identifying such agents in human remains. This contributes to reconstructing the historical disease burden—particularly from infections causing fevers, tuberculosis, plague, and syphilis—which likely played a major role in infant and child mortality[10].

This study presents an ancient case of *S. pyogenes*, offering a rare glimpse into the historical presence and genetic makeup of this important human pathogen, which still represents a modern thread linked to disease outbreaks. It expands our knowledge and approach to detecting pathogens that are thought to have contributed significantly to the burden of disease in ancient times. While this single case provides valuable evolutionary context, a broader dataset of ancient *S. pyogenes* genomes from both the Old World and the New World will be essential to reconstruct the pathogen's long-term evolutionary trajectory via *de-novo* assembly not just by comparison to a reference genome.

## Methods

### Ethics and inclusion statement
This research was conducted in accordance with ethical guidelines for the treatment of human remains[54]. The mummified human remains analysed in this study are curated by the National Museum of Archeology—MUNARQ in Bolivia. All necessary permits and authorizations for the study were obtained from the Bolivian Ministry of Cultures, Decolonization, and Depatriarchalization (AUTORIZACIÓN MDCyT—UDAM No. 017/2018). The mummified head is part of a historical collection composed of human remains and individuals recovered from diverse geographical regions. Based on current records and consultations, no specific Indigenous community has claimed affiliation with or recognized a direct connection to this individual.

### Sample collection
The MUNARQ in La Paz, houses the largest collection of mummified human remains and human skulls in Bolivia[55]. Minimally invasive sample collection was performed. We collected a tooth sample from a partially mummified head belonging to the anthropological collection (Sample ID: 2730/Museum ID: MUNARQ 2-87) (Fig. 1a).

### Burial context and palaeopathological overview
The mummified human remains housed in the MUNARQ were recovered from the *Chullpas*, or funerary towers, spread across the Bolivian Altiplano. During the Late Intermediate Period (1100–1450 AD), these individuals were interred within funerary bundles and placed inside these structures. While there are some accounts of anthropogenic mummification practices, most of the remains found in this region likely underwent a process of natural mummification over time.

The palaeopathological examination of individual ID:2730 suggests that this male individual (approx. 18–25 years-old) experienced intentional cranial modification. Dental assessment reveals dental chipping and localized periapical infections involving both mandibular first molars ($M_1$). A linear fracture is observed on the inferior portion of the occipital bone, extending posteriorly from the foramen magnum along the sagittal plane. Furthermore, post-mortem fractures of the cervical vertebrae, specifically involving the axis, show evidence of mechanical tension consistent with the deliberate removal of the cranium after death.

### DNA extraction and sequencing
All laboratory work was performed at the ancient DNA facilities of the Eurac Research, Institute for Mummy Studies in Bolzano, Italy. DNA

sampling preparation was conducted following the protocol developed specifically optimized for the recovery of ancient pathogens[56]. The procedure involved the transverse sectioning of the dental crown, followed by the mechanical drilling of the pulp chamber, which serves as a reservoir for endogenous pathogen DNA. This area is targeted because the highly vascularized pulp tissue historically trapped circulating pathogens, preserving their genetic material within the tooth structure. DNA extraction was performed on 50 mg of drilled tooth powder using EDTA-based demineralization followed by DNA precipitation using the linear polyacrylamide (LPA) precipitation method[57]. The extracted DNA was then converted into genomic DNA libraries following a protocol specific for ancient DNA[58]. The method focuses on "double-indexing" to maximize sample identification during high-throughput sequencing. It involves blunt-end repair, adapter ligation, and a specialized indexing PCR to create libraries ready for Illumina platforms. The prepared libraries were subjected to Illumina next-generation sequencing using Illumina HiSeq 2500.

### Sequencing quality controls
After demultiplexing of the reads, the sequencing adapters and low-quality reads were trimmed using fastp v0.23.1[59], allowing for a minimum base quality of 20. The quality-filtered reads were merged using fastp with a minimum overlap between forward and reverse reads of 10 bases.

### Human DNA analysis
For the analysis of human DNA, we mapped the merged reads against the reference human genome (build hg19), using BWA-aln v0.7.17[60] (using the following parameters: -l 1000 -n 0.04 -o 1 -e −1), then filtered for minimum mapping quality of 25 using SAMtools v1.17[61]. The mapping quality and coverage were then checked using Qualimap v2.2.2 d[62]. The ancient DNA damage pattern was checked using MapDamage 2 v2.2.1[63]. The mitochondrial DNA contamination was checked using Schmutzi v1.5.7[64]. The damaged sites of aDNA reads in the BAM file were then rescaled using MapDamage 2 and used to call a consensus sequence using the Schmutzi script "log2fasta", for mitochondrial haplogroup assignment using Haplogrep3 v2.4.0[65]. Molecular sex identification was performed from shotgun sequence data[20].

### Radiocarbon dates and stable Isotopic data
The remaining sample ID:2730 was sent for radiocarbon dating to the Department of Anthropology, AMS Radiocarbon Lab, Institute of Energy and the Environment, Penn State University, USA. The date was then used to assign the individual to a cultural period (Supplementary Data 1, Fig. S1). Radiocarbon dates were calibrated using the Oxcal software v4.4.4[66], applying the (SHCal20) for the Southern Hemisphere calibration curve[67]. Additionally, stable isotopic analysis of carbon ($\delta^{13}C$) and nitrogen ($\delta^{15}N$) in the dental sample was analysed to reconstruct the individual's diet, environmental conditions, and potential reliance on specific crops or animal resources during their lifetime (Supplementary Data 1).

### Microbial DNA analysis and metagenomic assembly
The general microbial taxonomic classification was carried out using MetaPhlAn 4 v4.0.6[68], with the database version of "June23". Then, to screen the other publicly available ancient metagenomic datasets, we used the tool AMDirT v1.4.5[69] to retrieve the raw fastq files from AMDir[22], which were analysed as described above.

Then, we used MEGAHIT v1.2.9[70] and metaSPAdes v3.15.5[71] to perform *de-novo* metagenomic assembly on the Bolivian sample, as well as other samples from the ancient metagenome directory that displayed the presence of *S. pyogenes*. For each sample, the resulting contigs were filtered for a minimum length of 1000 prior to using a multi-binning approach employing the metagenomic binners MetaBAT2 v2.12.1,

MaxBin2 v2.2.7[72], and CONCOCT v1.1.0[73]. The outputs of different binners were then combined using DAS Tool v1.1.6 to select the highest quality genomes[74]. To assess the assembly quality in terms of completeness and contamination, we used CheckM2 v1.0.2[75]. We further used Genome Database Taxonomy toolkit GTDBtk v2.1.1[76] for taxonomic assignment of the bins.

### Improving the quality of ancient Bolivian *S. pyogenes* genome
The initial binning process yielded two bins identified as *Streptococcus pyogenes*: one assembled with MEGAHIT and the other with metaSPAdes. To refine these bins, we used Kraken2 v2.17.1 (with the standard database[77]) to classify all contigs and removed any contigs not assigned to the genus Streptococcus. This decontamination step excluded five contigs in total (two were unclassified and three classified as *Proteus* spp.) Notably, CheckM2 quality metrics for the bins remained unchanged after this filtering. Following decontamination, the cleaned bins were merged and reassembled using the long-read assembler Flye v2.9.3[78] to maximize genome completeness and quality.

### Functional annotation and typing of the Bolivian *S. pyogenes* genome
To perform functional annotation of the ancient *S. pyogenes* genome, we used prokka v1.14.6[79], using the genome *Streptococcus pyogenes* M1 GAS (Strain: SF370; assembly ID: ASM4323122v1) as a reference. For the specific virulence factors, the annotated proteins were further compared against the NCBI non-redundant protein database using BLASTp v2.12.0, and the highest matches based on bitscore, query coverage, and sequence similarity were reported (Supplementary Data 10).

### Collection of modern comparative genomes
To perform a comprehensive phylogenetic and comparative analysis, we utilized EnteroBase (https://enterobase.warwick.ac.uk/), a platform for exploring genomic epidemiology[80]. Specifically, we downloaded genomes of *Streptococcus pyogenes* and other two closely related species, i.e., *Streptococcus dysgalactiae* and *Streptococcus canis*. In EnteroBase, Hierarchical Clustering (HC) groups bacterial isolates based on core-genome Multi-locus Sequence Typing (cgMLST) allelic profiles, with cluster levels (e.g., HC5, HC10, HC20, HC245) defined by the maximum number of allowable allelic differences, lower thresholds representing tighter, more recent genetic relationships. These clusters facilitate outbreak detection, epidemiological tracking, and population structure analysis by organizing isolates into hierarchically nested groups of increasing genetic diversity[81]. We opted to use the HC245 after exclusion of genomes with missing origin and isolation date in the metadata, which resulted in retrieving 149 genomes for *S. pyogenes*, 10 *S. dysgalactiae*, and 6 genomes for *S. canis*. A full list of genomes with metadata is available in Supplementary Data 14. These genomes were used further for phylogenetic placement, pangenome analysis, and functional comparison.

### Typing, antibiotic resistance gene, and virulence factor analysis
Multi-locus sequence types (MLST) and emm-types were determined from the Bolivian strain as well as the modern genomes using MLST v2.23.0 (https://github.com/tseemann/mlst) with the PubMLST database[82], and the emmtyper command line tool (https://github.com/MDU-PHL/emmtyper), respectively. For emmtyper, we used the following parameters to allow detection of both *emm* and *enn* proteins: "--percent-identity 75 --culling-limit 1 -f verbose --mismatch 35 --gap 4". The investigation of virulence factors and antibiotic resistance genes was carried out with ABRicate v1.0.0 (https://github.com/tseemann/abricate) tool by using the Virulence Factor Database (VFDB)[83], and the Comprehensive Antibiotic Resistance Database (CARD)[84] with 80% coverage ("--mincov") and 80% identity ("−minid") parameters.

## Phylogenetic and molecular clocking analysis

Then, gene prediction and annotation were carried out using the prokka pipeline[79]. To identify the core genome, we used Panaroo v1.5.2[85], a bioinformatics tool designed for bacterial pan-genome analysis. Panaroo identifies core genes (present in all strains), accessory genes (present in some but not all strains), and strain-specific genes. For this study, we set the similarity threshold at 95% for amino acid sequences to define the core genome. The core genes identified by Panaroo were aligned using MAFFT v7.525[86], a highly accurate multiple sequence alignment tool. The aligned sequences were then concatenated into a single alignment for phylogenetic analysis.

Divergence times have been estimated using BEAST.X. v1.10.5[87] and the BEAGLE library[88]. We preliminary tested between HKY and GTR replacement model and between a Constant and a Bayes skyGrid coalescent model by comparing marginal likelihoods: GTR + G was favored over HKY ($D = +66292$), and Constant was favored over Bayesian SkyGrid ($D = 1865$). According to model selection we therefore used the following priors for the main analyses: (1) GTR replacement model coupled with 4 categories of Gamma distribution; (2) A Constant Coalescent demographic/tree model; (3) The age of all samples for tip calibrations[89] by sampling uniformly from date uncertainty using as date the year of sample collection, an uncertainty of one years for all sample, and an uncertainty of 50 years for the Bolivian mummy (see suppl. Supplementary Data 14 for age of samples); (4) Strict clock with a normal prior rate distribution with mean $1.8 \times 10^{-6}$ and sd $7 \times 10^{-7}$: this allows covering between $4.3e^{-7}$ and $3.2e^{-6}$ at the 97.5% a range that is compatible with the rates of $1.8 \times 10^{-6}$ previously identified for *S. pyogenes*[90] and safely covers the posterior proposed for other *Streptococcus*, *S. dysgalactiae* ($5.8 \times 10^{-7}$) and *S. canis* ($8.6 \times 10^{-7}$) in respectively[91,92]. To test for outgroup effect, we repeated the analysis using a dataset composed of only *S. pyogenes*, and to test for taxon sampling, we repeated the analysis using a reduced set of 50 samples including both outgroups and *S. pyogenes*; for both datasets, we used the same combination of priors of the main analysis (GTR + G, Constant coalescent, all tip priors). All MCMCs were run for a minimum of 50 million generations until convergence. Convergence was inspected using Tracer by checking that all parameters had an ESS > 200 and setting a burn-in that maximized ESS. Analysis was run in double to assess further convergence, and trees were pulled from one of the two runs after burn-in using the HIPSTER method in TreeannotatorXv10. The resulting phylogenetic tree was visualized using iTOL v7 (Interactive Tree of Life), a platform for annotating and exploring phylogenies[93].

## Phage genome analysis

Further examination of all 29 assembled contigs revealed the gene composition (by contigs), including the presence of bacteriophages in contig 5 and in contig 26. Prophage predictions were made with PHASTEST[94], geNomad v1.7.4[95], VIBRANT v1.2.1[96], and PhageBoost v0.1.3[97]. Manual inspection of the prophage predictions revealed little difference between the prophage miners, and only the geNomad predictions were retained. Predictions were assessed with CheckV v1.0.3[98], and two near-complete prophages were retained. Annotation was first performed with Pharokka v1.7.5[99] with -g prodigal-gv[100,101] that uses tRNAscan-SE 2.0[102], Aragorn[103], CRT[104], PHROGs[105], VFDB[83], CARD[106], MMseqs2[107], PyHMMER[108], INPHARED[109] and MASH[110]. Subsequent annotation was performed with PHOLD v0.2.0[111] that implements Foldseek[112] and ProstT5[113], and Phynteny v0.1.12[114].

The two prophage predictions were compared to INPHARED (January, 2024)[109] using MASH v2.3[110]. Comparison of the prophages to their nearest hits revealed a seemingly missing section on both of the prophages. We then used MASH to compare the nearest hits to other contigs in the MAG, revealing four other contigs that were likely part of the two prophage genomes, however it remains unclear which of these additional contigs corresponds to which of the two prophage regions.

Prophage sequences were extracted from the modern genomes from Enterobase (BLAST reference database) using geNomad v1.7.4[95]. An all vs all comparison of prophages was made using MASH-dist to determine which reference *S. pyogenes* genomes contained similar prophages to those in the MAG. Similar prophages from bacterial genomes in the same clade as our MAG (Figs. 3 and S4) were extracted and de-replicated following Minimum Information about an Uncultivated Virus Genome (MIUViG) standards (95% ANI over 85% length) using blast v2.14.0+ alongside the anicalc.py and aniclust.py scripts described in the CheckV documentation[98]. The species representative reference prophages were compared to our prophage using Clinker v1.3.2[115].

## Analysis of functional modules in the pangenome

The resulting protein sequences (faa files) from prokka from each genome were used as input for further functional annotation. Functional annotation was performed using KofamScan v1.3.0[116] to assign KEGG Orthology (KO) identifiers to predicted genes. KofamScan searches were conducted against the KEGG profile database using a threshold score of 1 and output in detailed TSV format. KO assignments were made based on profile Hidden Markov Models (HMMs) with pre-defined score thresholds to ensure high-confidence annotations.

Assigned KO identifiers were mapped to KEGG modules using the official KEGG ko_to_module mapping file (accessed January 2025). For each sample, unique KO identifiers were extracted from KofamScan results and cross-referenced with the ko_to_module database to determine module presence. Module abundance was quantified as the count of unique KO identifiers belonging to each module within each sample.

## Statistical analysis and visualization

All statistical analyses and visualizations were performed in R version 4.5.1 using the tidyverse (v2.0.0) and pheatmap (v1.0.13) packages. Module count matrices were constructed with modules as rows and samples as columns, with values representing the number of unique KOs detected per module per sample. Modules present in fewer than 5 samples were excluded from downstream analysis to reduce noise.

To identify modules that differ significantly between the three *Streptococcus* species (*S. pyogenes*, *S. dysgalactiae*, and *S. canis*), we applied the Kruskal–Wallis rank sum test to each module independently. *P*-values were adjusted for multiple testing using the Benjamini-Hochberg false discovery rate (FDR) correction, with modules showing FDR < 0.001 considered statistically significant.

Module abundance patterns were visualized using hierarchical clustering heatmaps with Euclidean distance and complete linkage clustering. Count data were log10-transformed (log10(count + 1)) and row-scaled (z-score normalization) for visualization. Sample metadata, including species, geographic continent, and host type, were displayed as color-coded annotation bars.

## Reporting summary

Further information on research design is available in the Nature Portfolio Reporting Summary linked to this article.

# Data availability

Sequencing data and the assembled genome are available at the European Nucleotide Archive (ENA) under ENA: PRJEB91735. The sequencing reads are available at the Sequence Read Archive under accession ERR15308372 and the assembly under accession number GCA_982145515.1. Source data is provided with this article. Source data are provided with this paper.

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

## Acknowledgements

We thank the staff of the Museo Nacional de Arqueología—MUNARQ in La Paz, Bolivia for allowing access to the mummy collection (particularly Luis Castedo and Jelle Defrancq). This work was initially funded by the Marie Skłodowska-Curie Actions—Seal of Excellence Grant "MUMBO" Award Number: CUP: D55F20002570003; Provincia Autonoma di Bol-zano, grant legge 14. Department of Innovation, Research, and University of the Autonomous Province of Bolzano-South Tyrol (Italy). Furthermore, this research received funding from the Autonomous Province of Bol-zano/Bozen, within the framework of the call "Mobilità di ricercatrici e ricercatori", Project BOLD—Bolivian Mummies and Disease. "Paleoge-netic analysis of sixty Bolivian mummies provide insights into genetic history and health of pre-Columbian South Americans", (Decreto 13585/2023). Additional support was provided by the European Regional Development Fund 2014-2020 CALL-FESR 2017 Research and Innova-tion Autonomous Province of Bolzano South Tyrol Project: FESR1078—MummyLabs. We thank José Capriles from the Department of Anthro-pology, Penn State College, USA, and Brendan Culleton from the AMS Radiocarbon Lab, Institute of Energy and the Environment, Penn State University, USA for helping with the C14 and isotopic data analysis. We also thank Jasmin Niederkofler for her support in the wet lab analysis and Jérôme Zürcher for his help in the genome analysis. E.M.A. acknowl-edges support from the Biotechnology and Biological Sciences Research Council (BBSRC) BBSRC Institute Strategic Programme Food Microbiome and Health BB/X011054/1 and its constituent projects BBS/E/QU/230001B and BBS/E/QU/230001D, and by the BBSRC Institute Strategic Programme Microbes and Food Safety BB/X011011/1 and its constituent projects BBS/E/QU/230002 A, BBS/E/QU/230002B, and BBS/E/QU/230002 C. R.C. and E.M.A. are additionally funded through the Biotechnology and Biological Sciences Research Council (BBSRC) Grant Bacteriophages in Gut Health BB/W015706/1. We are grateful for the support of the Life Science Compute Cluster (LiSC) of the University of Vienna. The authors thank the Department of Innovation, Research, and University of the Autonomous Province of Bozen/Bolzano, Italy for covering the Open Access publication costs.

## Author contributions

Conceived and designed the experiments: F.M., M.S.S., and G.V. Per-formed the experiments: G.V., M.S.S., and F.M. Analysed the data: G.V., O.R.S., R.C., M.S.S., and F.M. Contributed reagents/materials/analysis tools: E.M.A. and A.Z. Wrote the paper: G.V., M.S.S., R.C., and F.M.

## Competing interests

The authors declare no competing interests.
