## [Transparent Peer Review file · Nature Communications]

An ancient genome of *Streptococcus pyogenes* from a pre-Columbian Bolivian mummy

Corresponding Author: Dr Guido Valverde

Version 0:

Reviewer comments:

Reviewer #1

(Remarks to the Author)

Noteworthy Results:

This work presents compelling findings for both the ancient DNA and modern microbiology/epidemiology communities. It successfully identifies *Streptococcus pyogenes* in a pre-Columbian context while providing a detailed genomic reconstruction of the ancient strain, complete with virulence factors and prophage content. The study offers valuable insights into disease dynamics among ancient Amerindian populations, including a plausible hypothesis about systemic infection or sepsis as a potential cause of death.

Significance and Originality:

The study represents a significant advancement as the first genomic evidence of *S. pyogenes* in pre-Columbian America, extending our knowledge of the pathogen's historical presence by centuries. It effectively bridges gaps in understanding the evolutionary history of this clinically important pathogen, while aligning with established literature on GAS virulence and phage-mediated evolution (e.g., Brouwer et al., 2023; McShan et al., 2019). The complete absence of prior ancient genomic data for *S. pyogenes* underscores the originality of these findings.

Support for Conclusions:

The authors' conclusions are well-supported through comprehensive analyses including metagenomic assembly, phylogenetic placement, virulence profiling, and prophage characterization. The data establish convincing connections between the ancient strain and modern lineages while demonstrating remarkable conservation of pathogenic mechanisms over time.

Flaws and Limitations:

While no major flaws were identified in the data analysis and the methodology meets field standards, certain aspects merit consideration. The phylogenetic analysis, while technically sound, requires cautious interpretation of geographic dispersal patterns due to the inherent weak phylogeographic signal in *S. pyogenes*. The shared prophages might reflect horizontal gene transfer rather than direct ancestral inheritance.

The mixed infection hypothesis presents a compelling case supported by metagenomic evidence of multiple contemporaneous pathogens. This approach proves particularly valuable for detecting ancient systemic infections that typically leave no skeletal markers. Future investigations incorporating biochemical markers, such as proteomic signatures of inflammation, could further validate such findings in cases of ancient sepsis.

The detection of *mreA*, *lmrP*, and *mef(E)* resistance genes in the ancient genome is intriguing, though the clinical relevance in this pre-antibiotic-era context remains uncertain without functional validation through phenotypic assays. Similarly, while the prophage reconstructions suggest functional potential, assembly ambiguities and reliance on computational predictions somewhat limit definitive conclusions about their activity *in vivo*.

Methodology and Reproducibility:

The methods are sound and meticulously detailed, fully meeting the expected standards for ancient DNA and metagenomic research. The complete workflow—from DNA extraction through phylogenetic analysis—is presented transparently with appropriate controls for contamination and damage patterns, ensuring reproducibility.

Final Assessment:

This study delivers unprecedented insights into the ancient genomic landscape of *S. pyogenes*, supported by robust methodology and high-quality data. While certain interpretations regarding phylogeography and prophage activity would benefit from additional caveats, the work undoubtedly establishes a new benchmark in pathogen paleogenomics. It will undoubtedly inspire and inform future research into the evolutionary history of human pathogens.

Minor comments:

In figure 1d, I would place a blue dot, to keep consistency, rather than colouring the whole country as blue. Figure S2 does not contain any black star as stated in the caption.

Reviewer #2

(Remarks to the Author)

The authors provide a high quality genome sequence of *S. pyogenes* from a Bolivian pre-Columbian mummified individual. I cannot comment on the accuracy of the radiocarbon dating as it is not within my area of expertise, yet the pathogen genome analyses seems robust where a 1.7Mb *S. pyogenes* draft genome is reported. Thank you for providing access to the raw data which appears of high quality and the assembly looks adequate (considering a <10x read coverage). The report is largely descriptive in line with the nature of the study, which may be enhanced with some additional core genome analyses to improve temporal evolutionary estimates. This genome provides a framework for more informed understanding of the architecture and evolutionary framework of the 'modern' *S. pyogenes* population. Comments:

1. While reporting virulence genes, is the putative location of these genes consistent with a modern *S. pyogenes* genome? Eg. Is the location of the emm gene within an archetypical Mga operon? And located in the same core genome location it is in modern genomes?
2. One element lacking is a report of the classic regulatory elements. Does the ancient genome have a classic complete covR/S regulator? The incorporation of other known regulatory genes would be of benefit.
3. The annotated map of the *pyogenes* genome in Figure 2a lacks some information. Perhaps a mauve alignment with a modern reference would help to contextualise the overall architecture. Why a question mark in the legend when reporting "biins"? The shaded heat map of phage mash distances is uninformative.
4. Where are the two intact prophages located? Please use the McShan et al nomenclature for this. When reporting the prophage elements in the text, please add which virulence genes are carried by which phage (and also in figure 3). Can the authors confirm whether the ssa gene (mentioned on page 8) was associated with a prophage (in line with modern genomes)?
5. *S. pyogenes* prophage typically are ordered by the integrase gene. In figure 3, it appears as though the assembly is disordered and the genetic makeup of the two prophage are very similar across several phage modules. This figure appears to be a pairwise alignment, yet this is not clear with the shading applied for the coloured phage modules. Suggest that the authors revise this figure to make the pairwise alignment more clear. The authors infer in their results that these assignments may be ambiguous. Can the assembly graphs be provided as supplementary to support their inferences. This confusion is reiterating in the final results paragraph, where the authors suggest degradation in these prophage and as such, may not represent "represent complete genomes with all necessary machinery for excision and replication" as inferred in the preceding paragraphs.
6. Could the authors provide some comparison on the overall gene functional categories in the ancient genome relative to modern genomes? I.e. are there any genes in the ancient genome that is NOT found in modern genomes (pan-genome analyses)? Was the number and type of gene function categories the same? Eg metabolic genes. eggNOG or equivalent would enable this functional categorisation of the strain relative to modern references. A reader may find this comparison interesting considering that diet changes that have occurred in the human host over time (or as the authors suggest may in fact be multi-host).
7. Generally in ancient genome studies, the presence of an ancient phylogenetic root facilitate temporal style phylogenetic analysis. Perhaps the authors would consider Bayesian analysis to improve estimates regarding ancestral evolution of *S. pyogenes* (core genome)? Eg. Within the "HC10 clusters" (unclear to this reviewer what this cluster designation refers to) section where "evolutionary placement" phylogenetic inferences were undertaken.
8. In the discussion, the authors extrapolate their findings to infer invasive potential "These findings also suggest that it would belong to an invasive group A Streptococcus (GAS) lineage capable of causing severe infections in ancient human populations." This is a loose inference, and should be tempered.

Minor comments:

Lowercase emm when referring to the gene (even at the start of a sentence). Eg. Emm137 should be emm137

Please italicise "Streptococcus" throughout.

Please double check references. There appears to be some duplication (eg 39 and 5).

Reviewer #3

(Remarks to the Author)

This is an extremely interesting data set. My primary criticism (points #1-#5) is that the sequence data has not been analyzed to sufficient depth. There are some key questions that remain unanswered - however they could be readily addressed, and such an analysis is likely to elevate the impact and value of the study. A second concern lies in speculations on data interpretation (points #6-#9).

Overall, I find some of the summations of the *S. pyogenes* and related literature to be simplified and/or out of context (and probably unnecessary). The manuscript could likely be strengthened by a more focused approach that includes a more thorough evolutionary analysis of this important genome sequence. Specific comments are listed below:

1. One key piece of missing analysis is an in-depth comparison of the Bolivian isolate genome to *Streptococcus equisimilis* subspecies *dysgalactiae* (SDSE) – SDSE are the closest known genetic relatives of *S. pyogenes*. SDSE and *S. pyogenes* display an extensive history of horizontal gene exchange and share a very large number of virulence genes (emm, FCT-9 pili, streptolysins, streptokinase, many others), core genes (including MLST overlap) and mobile genetic elements (phage genes, R-genes). For example, please see Xie et al (Nature Comm, 2024), among several other publications on *S. pyogenes*-SDSE gene flow. A deeper analysis of the Bolivian isolate versus closely related groups C and G strep genomes seems warranted. *Strep. agalactiae* was chosen for comparison as a related species, however, it is more genetically distant.
2. A second key piece of missing analysis: Most *S. pyogenes* strains have an emm gene plus 1 or 2 emm-like (emmL) genes; thus, one cannot necessarily assign emm-type based on a sequence hit alone. Evidence should be provided showing that the so-called 'emm137' gene corresponds to the correct genome map position for the type-specific emm locus - and not the downstream (paralogous) emmL/enn locus. Most emm genes have a history of horizontal transfer and recombination, and most emm types have been recovered from multiple divergent genetic backgrounds. The (rare) emm137 Fiji strain cited in reference 29 has a distinct FCT-region (not FCT-9) and therefore, does not appear share a recent/direct common ancestor with the Bolivian strain.
3. Related to #2: An in-depth and granular Blast analysis that includes the entirety of the emm-region – spanning from the Mga locus to the C5a peptidase locus (whether assembled or in fragments) – might address the emm/emmL '137' issue, as well as provide key evolutionary insights since the emm-region of the genome is subject to strong diversifying selection. It may be insightful to know which sub-portions of the emm-region and emm/emmL gene product functional domains display high homology to contemporary *S. pyogenes* genes, and which parts are unique and non-extant.
4. A map of the FCT region genes might also be a nice addition. It is stated that the Bolivian isolate displays the FCT-9 region form. Is the pilB gene highly homologous to any contemporary alleles? How about other FCT-9 region genes? Several FCT-9 loci of *S. pyogenes* are shared with group C and G streptococci.
5. One of the benefits of a dated ancient genome is that the data can be used to refine the 'molecular clock' and mutation rates. Although often such analyses yield soft conclusions, a thoughtful attempt here might be worthwhile. A deeper analysis of select core genes, for their sequence divergence with modern-day strains, may also provide interesting new evolutionary insights. Additional data supporting the novelty of the ancient isolate can help to minimize any lingering doubts that it is a modern-day contaminant.
6. Descriptive annotations derived from bioinformatics software do not by themselves equate to biological function. There are several instances in the manuscript where the authors delve into commentaries on annotated genes as virulence- or resistance-associated - yet those ascribed gene functions in this species (*S. pyogenes*) are not necessarily experimentally proven phenotypes. Unless biological functions are supported by (accurate) citations, the discussions seem way too speculative; this concern extends to the Abstract.
7. What is meant by "invasive" lineage? Most strains/emm-types of *S. pyogenes* can (more or less) cause invasive disease.
8. The commentary on a systemic infection, with a possible mixed infection, seems weak and not well-connected to current clinical knowledge. The 'co-isolates' *Clostridium tetani* and *botulinum* (and *Morganella*) are primarily environmental (soil; and the clostridia form endospores, perhaps preserving their DNA); it is extremely rare for these species to be present in the bloodstream (unlike *S. pyogenes*). Perhaps the person had sepsis due to *S. pyogenes* or perhaps they had a polymicrobial tooth abscess (although a mixed infection would more likely include anaerobes from the oral microbiota), or maybe the cause of death was non-infectious – one can only speculate.
9. The discussion on the population structure of *S. pyogenes* seems a bit out of place and unclear.
10. There are a few reports of natural recovery of *S. pyogenes* from non-human primates, although there is no strong evidence that this bacterium is a pathogen of consequence (thereby 'human-specific'). The authors may wish to consider their discussion within this context.
11. The references lack journal titles. Some citations do not provide the evidentiary support that the authors claim.

12. Line numbers would be helpful.

Reviewer #4

(Remarks to the Author)

In this manuscript, Valverde and colleagues present the ancient DNA analysis of a tooth sample from a pre-Columbian era mummy from Bolivia. The highlight of the study is the recovery and characterization of a near-complete genome of a pathogen, *Streptococcus pyogenes*.

The paper is generally well-written. The analyses related to the phylogeny and genomic architecture of the *S. pyogenes* strain are well done and interesting to read. However, the authors need to do a better job of convincing the readers regarding the ancient origin of the pathogens presented in this paper. The following comments need to be addressed before I can recommend the manuscript for publication:

The authors state "Further microbial profiling using MetaPhlan4 displayed unusual high abundances of different human pathogenic bacterial species - *Clostridium tetani* (37.5%), *Morganella morganii* (29.5%), *Streptococcus pyogenes* (11.6%), *Clostridium botulinum* (6.7%), *Asaccharospora irregularis* (6.4%), and *Paeniclostridium sordellii*". Typically when one is analyzing ancient pathogen DNA from a tooth sample, the pathogen's DNA is very low in abundance as compared to that from oral microbes. The profile given here strikes me as very unusual. Given the high abundance of these six taxa in the sample, one might hypothesize that these are contaminants that have leached into the tooth from the burial environment. This possibility needs to be discussed and the authors need to make a better case for why they think these are ancient taxa, endogenous to the tooth sample, and why their presence in this sample indicates that the individual suffered from an infection by these pathogens. For e.g. *Clostridium tetani* and *C. botulinum* can both be found in the soil. *Asaccharospora irregularis* was formerly considered to be a *Clostridium*, so it is likely found in the soil too, but I can find no reference of it being a human pathogen.

From Table 1, it seems the *S. pyogenes* DNA shows 1.2% and 1.7% damage at the 5' and 3' ends, respectively, whereas the human DNA from the same tooth shows 3.7% and 5.1% damage. The authors need to discuss this discrepancy between the amount of damage for the host and pathogen DNA.

The argument for mixed infection seems weak to me, especially since I find it less likely that the *Clostridium* species are endogenous to the tooth sample. In any case, the authors will need to give damage patterns for the other pathogens in order to support an argument for mixed infection.

It is disappointing that there is no discussion of the burial context for this individual. Furthermore, was any paleopathological analysis done that suggests the presence of any particular disease(s)? Also, please specify which tooth was sampled.

The DNA extraction and sequencing section need more details. Please briefly describe the extraction and library preparation protocol used.

Version 1:

Reviewer comments:

Reviewer #1

(Remarks to the Author)

I thank the authors for their careful and thorough revision. My concerns have been adequately addressed, and the manuscript has been substantially strengthened. I have no further comments and support publication in its current form.

Reviewer #2

(Remarks to the Author)

The authors have made substantial alternations to the manuscript including new analyses which address the major concerns/queries. The revised version has substantially improved and the authors should be commended for their diligence. The supplementary material is important for validation.

While the raw data looks to have been deposited, it is strongly recommended that draft assemblies generated be made public available.

Reviewer #3

(Remarks to the Author)

The extensive additional analyses and thoughtful responses to reviewer concerns are greatly appreciated. The revised manuscript seems much better focused and importantly, seems well-poised to have a high impact on the *S. pyogenes* field.

I have a few suggestions for further clarifications:

1. Lines 79-80. "Like *S. pyogenes*, SDSE commonly expresses Lancefield group C or G antigens....." Incorrect as written: *S. pyogenes* express Lancefield group A carbohydrate.

Also, if the authors know that their isolate has high homology to the group A carbohydrate biosynthesis genes (*gac* genes), it may be of added value to include that point in their analysis of other key genes (Results section)

2. Line 83: Omit the "Thereby" – Exchange of many [virulence] genes is by mechanisms unknown (i.e., not yet proven for mobile genetic elements as the vehicle, except for [largely] pyrogenic exotoxins via transduction)]. Also lines 85-87.

3. Line 253 (and related Discussion): animal infection models for pneumococcus (not *S. pyo*)? Ref. 35

4. Line 276-paragraph (and related Discussion): Is the *ImrP* gene product actually known to confer antibiotic resistance in this species (*S. pyo*)? Citation? Also, a functional 'mef[E]' is far more typical for other streptococcal species. The ideas in this section seem a bit overstated.

5. Lines 396-399: It is not all that uncommon for modern strains to lack phage-associated virulence (superantigen) genes. So perhaps this idea is not quite correct as presented.

6. Line 400: "*S. pyogenes* is predominantly a clonal pathogen" (ref. 43 cited). Authors should probably clarify or soften. Global strains of *S. pyogenes* taken as a group are nearly as 'recombinogenic' as pneumococci via numerous measures. There exist some strongly 'clonal' lineages within *S. pyogenes* (e.g., M1T1) and many lineages/clusters (phylogroups) that are more diverse.

Reviewer #4

(Remarks to the Author)

The manuscript by Valverde and colleagues is much improved following the previous round of revision. I am satisfied with their response to the concerns I had previously raised. Below I have noted some additional (mostly minor) concerns that need to be addressed.

Lines 319-324: The authors infer that the timing of the origin of *S. pyogenes* is the same as the timing of its split from *S. dysgalactiae* i.e. 137,264 years. I don't quite agree with this inference; in my opinion, the timing of the MRCA of all *S. pyogenes* strains (ancient and modern), which is estimated to be 10,270 years, is what one should consider as the origin of this species. If the authors disagree with this, I encourage them to include a sentence stating why they think using the stem age (time at which *S. pyogenes* diverged from its sister taxon), as opposed to the crown age (time of MRCA for all *S. pyogenes*), would be appropriate to consider as the time at which this species originated.

Line 532: Please specify which bwa algorithm was used – mem or aln? Were default parameters used? If using aln, were recommended aDNA parameters used (such as disabling the seed, changing mapping stringency to allow reads with damaged nucleotides to map)? If yes, please specify the parameter values.

Line 533: With regards to Qualimap, do you mean to say coverage instead of mapping quality?

Line 536: It is unclear how the consensus sequence for the human DNA was called, since this cannot be done using either MapDamage2 or Haplogrep. Did you mean to say you rescaled the damaged sites using MapDamage2 and then called the consensus using schmutzi?

Line 631: Correct "EES" to "ESS". Also, specify what was considered as a "satisfactory" ESS value.

Apart from the above minor concerns, the manuscript needs to be thoroughly proof-read to correct instances of awkward phrasing and lingering typos. The instances I could find are given below.

Line 30: Missing period at end of sentence. Add "the" before "European colonial expansion".

Line 37: Use "the" in place of "an".

Line 56: Add comma after "health".

Lines 96, 124: Add "the" before "pre-Columbian".

Line 104: Missing period after the "S" of *Streptococcus*.

Line 105: Rephrase to "... and calibrate the pathogen's phylogeny..."

Line 109: Rephrase to "...tooth sample from a pre-Columbian..."

Line 129: Rephrase to "...environments with dry conditions which have been previously shown to be less prone to DNA degradation."

Line 130: Remove space between MetaPhlAn and 4.

Line 138: Rephrase to "...to this species to understand..."

Line 140-142: Rephrase to "...which revealed the presence of *S. pyogenes* in human samples from different time periods (ranging from 4000 to 200 years BP) in Europe and Africa, as well as in gorilla samples from museum collections"

Line 144: I recommend using the word "samples" instead of the word "findings".

Line 164: Rephrase to "The completeness of the assembled *S. pyogenes* genome was 99.98%.."

Table 1: Needs to be properly formatted with respect to gray and white cell colors.

Line 179: Redundant "typing" after MLST.

Line 195: Remove the word "situation". I recommend using "currently available" instead of "current available".

Line 197: Rephrase to "...*pyogenes* strain to the modern emm type 46 (79% nucleotide identity)"

Line 234: Rephrase to "In addition to the surface-associated M- and T-proteins, the ancient *S. pyogenes* strain holds..."

Line 252: Change to "...pathogenicity in animal infection..."

Line 255: Specify here what you mean by "our prophages". I recommend saying "however, the prophages in the ancient strain...".

Line 290: Rephrase to "to be completely absent in other modern GAS genomes as well".

Line 297, 311, 312, 314, and elsewhere: Do not capitalize the word "gorilla".

Line 317: Remove the word "ancestral". It is redundant in that sentence.

Line 322 – 324: Rephrase to "The split of the Bolivian strain from all other *S. pyogenes* (divergence of known *S. pyogenes*) was estimated to be 10,270 years ago (95% HPD: 9314 to 11,218 cal yr BP), and a diversification of most modern *S. pyogenes* in the past 5,500 years of human history."

Line 370: Missing period after the word "infections".

Line 476: I recommend saying tuberculosis, instead of TB, as it is the first (and only) time the disease is mentioned in the manuscript.

Line 503: Remove comma after 2730.

Line 534 and 536: Remove space between MapDamage and 2.

Line 559: Give the full form of GTDBtk.

Line 579: This line is missing a verb after "we" for e.g. downloaded or acquired.

Lines 622-625: Please ensure that numbers that need to be shown as superscript are correctly formatted. For eg. 1.8 10⁻⁶ needs to be corrected to 1.8x10⁻⁷ or given as superscript.

General response to the Editor and Reviewers

We sincerely thank the Editor and all four reviewers for their careful evaluation of our manuscript and for the insightful, constructive comments. We greatly appreciate the time and expertise invested in these reviews. The feedback has been extremely valuable and has helped us to substantially improve the clarity, rigor, and scope of the study.

In response to the reviewers' main concerns, we have extensively revised and restructured the manuscript. In particular, we have refined our interpretations to avoid overstatement, expanded and strengthened the evolutionary analyses, and focused the functional discussion on the most robust and well-supported genomic features of *Streptococcus pyogenes*.

The major changes implemented in the revised manuscript include the following:

- 1) We have toned down the mixed-infection hypothesis, limiting our interpretation to reporting the presence of additional microbial taxa detected in the metagenomic data, and clarifying the uncertainties regarding their origin and biological relevance.
- 2) We have redone the phylogenetic analyses, incorporating the closely related species *Streptococcus dysgalactiae* and *Streptococcus canis*, and applied molecular clock approaches to estimate divergence times and better contextualize the evolutionary placement of the ancient *S. pyogenes* genome.
- 3) We have restricted the virulence analysis to the most prominent and well-documented virulence determinants of *S. pyogenes*, supported by established literature, and removed or downplayed more speculative functional interpretations.
- 4) We have toned down the prophage analyses, including simplifying the discussion of prophage similarity and reducing emphasis on MASH distance-based comparisons, while clarifying uncertainties related to assembly completeness and functional inference.

In addition, we have addressed numerous specific comments regarding genome architecture, regulatory elements, comparative genomics, figure presentation, terminology, formatting, and clarity of methods and discussion. All reviewer comments are addressed point by point, with references to specific line numbers where changes have been made in the revised manuscript. Instead of highlighting the changes with colours in the main manuscript we decided to copy the text changes into the respective reviewer response sections. Our responses are in blue font.

We hope that these substantial revisions adequately address the reviewers' concerns and strengthen the manuscript. We are grateful for the opportunity to revise our work and look forward to your further evaluation.

Reviewer #1 (Remarks to the Author):

Noteworthy Results:

This work presents compelling findings for both the ancient DNA and modern microbiology/epidemiology communities. It successfully identifies *Streptococcus pyogenes* in a pre-Columbian context while providing a detailed genomic reconstruction of the ancient strain, complete with virulence factors and prophage content. The study offers valuable insights into disease dynamics among ancient Amerindian populations, including a plausible hypothesis about systemic infection or sepsis as a potential cause of death.

Significance and Originality:

The study represents a significant advancement as the first genomic evidence of *S. pyogenes* in pre-Columbian America, extending our knowledge of the pathogen's historical presence by centuries. It effectively bridges gaps in understanding the evolutionary history of this clinically important pathogen, while aligning with established literature on GAS virulence and phage-mediated evolution (e.g., Brouwer et al., 2023; McShan et al., 2019). The complete absence of prior ancient genomic data for *S. pyogenes* underscores the originality of these findings.

Support for Conclusions:

The authors' conclusions are well-supported through comprehensive analyses including metagenomic assembly, phylogenetic placement, virulence profiling, and prophage characterization. The data establish convincing connections between the ancient strain and modern lineages while demonstrating remarkable conservation of pathogenic mechanisms over time.

Thanks for positive assessment

Flaws and Limitations:

While no major flaws were identified in the data analysis and the methodology meets field standards, certain aspects merit consideration. The phylogenetic analysis, while technically sound, requires cautious interpretation of geographic dispersal patterns due to the inherent weak phylogeographic signal in *S. pyogenes*. The shared prophages might reflect horizontal gene transfer rather than direct ancestral inheritance.

In response to Reviewer #1 and to address the concern of the other reviewers too, we conducted a Bayesian phylogenetic analysis including closely related species (*Streptococcus dysgalactiae* and *Streptococcus canis*) and applied molecular clock models to estimate divergence times and assess temporal signal in the core genome. We created a new "Figure 3" including the Bayesian phylogenetic analysis + functional analysis employing KEGG modules to compare the three different *Streptococcus* species.

In the revised phylogenetic analyses, after including the outgroups, the ancient Bolivian *Streptococcus pyogenes* genome (Bolivia 2730) falls in a basal position relative to the diversity of modern *S. pyogenes*. In both maximum-likelihood and Bayesian molecular clock frameworks, the ancient genome branches off prior to the radiation of contemporary strains, consistent with an early-diverging lineage within the species.

Additionally, we removed the MASH distances of the phages and confined the phage description on the annotation comparison with closely related species without reference to the phylogenetic relationship. Please refer to Figure 2b and Figure S4.

The mixed infection hypothesis presents a compelling case supported by metagenomic evidence of multiple contemporaneous pathogens. This approach proves particularly valuable for detecting ancient systemic infections that typically leave no skeletal markers. Future investigations incorporating biochemical markers, such as proteomic signatures of inflammation, could further validate such findings in cases of ancient sepsis.

We removed the mixed infection possibility from the results and from the discussion and kept the detected list of microbes/pathogens without stressing the mixed infection.

The detection of *mreA*, *ImrP*, and *mef(E)* resistance genes in the ancient genome is intriguing, though the clinical relevance in this pre-antibiotic-era context remains uncertain without functional validation through phenotypic assays. Similarly, while the prophage reconstructions suggest functional potential, assembly ambiguities and reliance on computational predictions somewhat limit definitive conclusions about their activity in vivo.

We thank the reviewer for this important clarification. We agree that the clinical relevance of these genes in a pre-antibiotic-era context cannot be inferred without functional validation. In the revised manuscript, we have therefore tempered our interpretation and now emphasize that *ImrP*, and *mef(E)* encode membrane-associated efflux systems with broad substrate specificity, which are not exclusively linked to antibiotic resistance. Such transporters are known to play wider ecological roles, including the export of toxic metabolites, host-derived antimicrobial compounds, and other environmental stressors, and may contribute to bacterial fitness independently of clinical antibiotic exposure.

Accordingly, we have revised the Discussion section to avoid implying phenotypic resistance and now frame these genes as components of ancestral stress-response and detoxification mechanisms that were later used for antibiotic resistance in modern clinical settings.

Here is the current text in the discussion section: “*The detection of macrolide efflux pump genes ImrP, and mef(E) in the ancient S. pyogenes genome provides compelling evidence for the long-standing presence of antibiotic resistance determinants prior to the modern clinical use of macrolides. These genes are commonly found in contemporary clinical isolates and are often associated with mobile genetic elements, suggesting that horizontal gene transfer may have played a role in their ancient dissemination. The high sequence identity and broad coverage observed indicate that these genes were likely functional at the time, supporting the hypothesis that resistance traits were already circulating in microbial communities before the antibiotic era (D’Costa et al. 2011). This finding supports the idea that resistance genes are ancient and naturally present in microbes, shaped by long-term ecological competition rather than just modern antibiotic use.*”

Methodology and Reproducibility:

The methods are sound and meticulously detailed, fully meeting the expected standards for ancient DNA and metagenomic research. The complete workflow—from DNA extraction through phylogenetic analysis—is presented transparently with appropriate controls for contamination and damage patterns, ensuring reproducibility.

Final Assessment:

This study delivers unprecedented insights into the ancient genomic landscape of *S. pyogenes*, supported by robust methodology and high-quality data. While certain interpretations regarding phylogeography and prophage activity would benefit from additional caveats, the work undoubtedly establishes a new benchmark in pathogen paleogenomics. It will undoubtedly inspire and inform future research into the evolutionary history of human pathogens.

Minor comments:

In figure 1d, I would place a blue dot, to keep consistency, rather than colouring the whole country as blue.

We adjusted the colour to the heatmap according to the sample age and kept the label coloured in blue in all figures.

Figure S2 does not contain any black star as stated in the caption.

The figure is updated.

Reviewer #2 (Remarks to the Author):

The authors provide a high quality genome sequence of *S. pyogenes* from a Bolivian pre-Columbian mummified individual. I cannot comment on the accuracy of the radiocarbon dating as it is not within my area of expertise, yet the pathogen genome analyses seems robust where a 1.7Mb *S. pyogenes* draft genome is reported. Thank you for providing access to the raw data which appears of high quality and the assembly looks adequate (considering a <10x read coverage). The report is largely descriptive in line with the nature of the study, which may be enhanced with some additional core genome analyses to improve temporal evolutionary estimates. This genome provides a framework for more informed understanding of the architecture and evolutionary framework of the 'modern' *S. pyogenes* population.

Comments:

1. While reporting virulence genes, is the putative location of these genes consistent with a modern *S. pyogenes* genome? Eg. Is the location of the emm gene within an archetypical Mga operon? And located in the same core genome location it is in modern genomes?

In the previous version of the manuscript, we used and reported the assembly version we retrieved from using the metagenomic assembler MEGAHIT. However, and to address the important question of the reviewer, we opted to try another assembly method, i.e. metaSPAdes. We realized improvement of some parameters and contigs lengths, therefore we merged the two assemblies using the long-read assembler Flye to achieve the best version of the ancient genome. We included comparative statistics in the new version of the manuscript and accordingly modified Table 1.

	Name	MEGAHIT	metaSPAdes	Flye
Assembly Quality	Completeness (%)	99.98	99.99	99.98
	Contamination (%)	0.28	0.29	0.28
Assembly Statistics	Total length (bp)	1707330	1695810	1703496
	Total length (>= 25000 bp)	1585200	1483076	1620257
	Total length (>= 50000 bp)	1263868	1163145	1336590
	Number of contigs	37	38	29
	Largest contig (bp)	250506	208929	250505
	N50 (L50)	70736 (8)	85333 (8)	99718 (7)
	N90 (L90)	26863 (24)	21758 (23)	29130 (19)
Gene Content	Coding Density	0.869	0.873	0.87
	GC Content	0.38	0.38	0.38
	Total Coding Sequences	1637	1631	1630
	CRISPR	1 - with 4 spacers	1 - with 12 spacers	1 - with 12 spacers
	tRNA	38	22	24
	rRNA	3 (16S, 23S, and 5S)	0	3 (16S, 23S, and 5S)

For the genome analysis, we used the combined version of the assembly (Flye) and here we refer to the positions according to its coordinates.

Regarding the mga regulon, we found the components of the mga regulon split into two different contigs: contig 17 includes the mga gene + partial emm and contig 13 includes enn + scpA. We confirmed homology of the enn and the partial emm using BLASTp as presented now in Table S10 and in Figure 2a.

And in Table S10:

contig	start	end	strand	coverage	identity	accession	gene	ID
contig_17	1347	2939	-	99	98.48	GEE0519.1	virulence factor transcriptional regulator Mga	mga
contig_17			-				Partial M-protein	emm
contig_13	190	1296	+	100	96.74	WP_136302769.1	M-related protein Enn	enn
contig_13	1665	5168	+	100	98.46	WP_326656969.1	C5a peptidase ScpA/B	ScpA

And for the partial emm of contig 17, we manually extracted this gene using BLASTx search and performed multiple sequence alignment against the closest hit and 3D alignments to confirm the results (Please refer to Figure S3).

Overall, the *mga* pattern we have in the ancient genome is *mga-emm-enn-scpA*. And we added the description of this *mga* regulon in the main text as follows:

“The emm gene is located in the mga regulon with genes encoding additional M-like proteins such as mrp and enn, and scpA encoding a C5a peptidase (Hondorp & McIver 2007). In the assembled Bolivian genome, the mga regulon genes are distributed to two different contigs, contig 17 that carries the transcriptional activator gene mga and a partial emm gene, and contig 13 that encodes the emm-like gene enn and the C5a peptidase gene scpA (Figure 2b). Considering the size range of 220 to 513 amino acids in modern M protein sequences (Frost et al. 2020), the emm gene of the ancient Bolivian strain is nearly complete (1161 bp, 387 aa in length) lacking only the 3′ gene part that encodes the D-repeat region of the M protein (Figure S3). This gene part could not be assembled with the current available data situation and assembly strategy. Nevertheless, emm typing targeting the 5′ region of the gene was possible and assigned the ancient S. pyogenes strain closest (79% nucleotide identity) to the modern emm type 46 (see Figure S4, Table S7). However, based on the currently existing typing criteria, the ancient strain should be assigned to a unique emm type, having less than 92% nucleotide identity over the first 90 bases encoding the mature M protein to any other emm type (Bessen et al. 2018a). Next, we extended our strain typing to the emm pattern analysis that groups emm types into functional patterns (A–C, D, E) that correlate with tissue tropism, ecological niche, and pathogenic behaviour (Bessen et al. 2022). Based on the chromosomal arrangements of emm and emm-like genes (mga-emm-enn-scpA, Figure 2b) and their subfamily (SF) forms (enn SF-1), the ancient Bolivian strain can be associated with the emm pattern group A-C, being most likely a S. pyogenes emm pattern B strain. The analysis of a global collection of group A Streptococcus revealed that majority of modern strains belong to emm pattern groups D and E, whereas emm pattern B or C strains are nowadays only rarely observed (Frost et al. 2020) (Table S8). Interestingly, the modern strains displaying highest sequence identity to the partial Bolivian emm gene and that share a similar configuration of the mga regulon mostly come from the Americas or Oceania (Table S8 and S9, Figure S3).”

2. One element lacking is a report of the classic regulatory elements. Does the ancient genome have a classic complete *covR/S* regulator? The incorporation of other known regulatory genes would be of benefit.

Yes, the ancient Bolivian genome contains the *covR/S* regulator in addition to other 10 different two-component system regulators. In the new version of the manuscript, we included a new supplementary table (Table S13) listing all of them.

Two-component system	Spy locus	Gene coordinates Bolivian strain	Function
ciaRH	spy1237/6	contig_8: 55834 to 57811	Metabolism, and stress responses
covRS	spy0336/7	contig_20: 70672 to 72866	Global regulation
fasBCA	spy0242/4/5	contig_1: 100186 to 103556	Virulence
irr/ihk	spy2027/6	contig_19: 25059 to 27097	Immunity evasion
liaFSR	spy1623/2/1	contig_3: 92225 to 94552	Immunity evasion
salkR	spy1908/9	ND	Antimicrobial peptide management
silAB	N/A	ND	Quorum sensing
sptRS	Spy0874/5	contig_5: 151596 to 153505	Virulence
spy1061/2		contig_12: 10486 to 12924	Activation of mannose/fructose-PTS system
srtRS	spy1081/2	ND	Antimicrobial peptide management
maekR	spy1107/6	contig_22: 1610 to 3797	Malate metabolism
spy1556/3		contig_7: 97026 to 99496	Global regulation
trxTSR	spy1589/8/7	contig_2: 20341 to 24144	Activation of mga regulon
vicRK	spy0528/9	contig_24: 12001 to 14056	Essential for growth, nutrition management

And in the main text, we added the following paragraph to the results:

“The expression of the above-mentioned virulence genes, the modulation of antibiotic resistance and stress-response genes, and the control of prophage and other mobile genetic element genes is

mediated by specific two-component regulatory systems (TCS) (Buckley et al. 2018). Group A *S. pyogenes* possesses up to 14 TCS to adapt the bacterial behaviour to host and environmental conditions. The ancient Bolivian strain encodes for 11 TCS including the classic complete covR/S TCS that acts as a global control for many virulence genes (**Table S13**). The three TCS *salkR*, *silAB*, and *srtRS* that are missing in the ancient strain are described to be completely absent also in other modern GAS genomes (Buckley et al. 2018)."

3. The annotated map of the pyogenes genome in Figure 2a lacks some information. Perhaps a mauve alignment with a modern reference would help to contextualise the overall architecture. Why a question mark in the legend when reporting "biins"? The shaded heat map of phage mash distances is uninformative.

The bins was a typo, and the figure is modified now.

In the new Figure 2a, we report the gene clusters/operon of the main regulatory/virulence genes which can help to contextualise the ancient strain. Namely, we report the gene clusters of FCT regions, *mga* regulon, Streptolysin S (*sag* operon), and *hasABC* genes.

In the results section, we described the arrangements of the *mga* and FCT regions in a way that can textualize them. The new text added is as follows:

"To systematically and functionally assign the ancient S. pyogenes strain we applied multi-locus sequence (MLST), emm, and pilin typing schemes that are commonly used to classify modern S. pyogenes strains (Bessen et al. 2018b; Bessen et al. 2022). MLST typing of seven housekeeping genes assigned the ancient strain to a distinct sequence profile, that has no direct modern equivalents and possibly represents an ancestral lineage of contemporary strains (Table S6). The most similar modern S. pyogenes isolates come from remote aboriginal island communities in tropical Australia sharing three out seven MLST alleles with the ancient strain (McGregor et al. 2004). Additional S. pyogenes typing schemes target the emm gene encoding the surface-exposed streptococcal M protein, and pilin adhesin and backbone genes (Bessen et al. 2024). Both M proteins and pili are key virulence factors and targets of host protective immunity (Walker et al. 2014). The emm gene is located in the mga regulon with genes encoding additional M-like proteins such as mrp and enn, and scpA encoding a C5a peptidase (Hondorp & Mclver 2007). In the assembled Bolivian genome, the mga regulon genes are distributed to two different contigs, contig 17 that carries the transcriptional activator gene mga and a partial emm gene, and contig 13 that encodes the emm-like gene enn and the C5a peptidase gene scpA (Figure 2b). Considering the size range of 220 to 513 amino acids in modern M protein sequences (Frost et al. 2020), the emm gene of the ancient Bolivian strain is nearly complete (1161 bp, 387 aa in length) lacking only the 3' gene part that encodes the D-repeat region of the M protein (Figure S3). This gene part could not be assembled with the current available data situation and assembly strategy. Nevertheless, emm typing targeting the 5' region of the gene was possible and assigned the ancient S. pyogenes strain closest (79% nucleotide identity) to the modern emm type 46 (see Figure S4, Table S7). However, based on the currently existing typing criteria, the ancient strain should be assigned to a unique emm type, having less than 92% nucleotide identity over the first 90 bases encoding the mature M protein to any other emm type (Bessen et al. 2018a). Next, we extended our strain typing to the emm pattern analysis that groups emm types into functional patterns (A–C, D, E) that correlate with tissue tropism, ecological niche, and pathogenic behaviour (Bessen et al. 2022). Based on the chromosomal arrangements of emm and emm-like genes (mga-emm-enn-scpA, Figure 2b) and their subfamily (SF) forms (enn SF-1), the ancient Bolivian strain can be associated with the emm pattern group A-C, being most likely a S. pyogenes emm pattern B strain. The analysis of a global collection of group A Streptococcus revealed that majority of modern strains belong to emm pattern groups D and E, whereas emm pattern B or C strains are nowadays only rarely observed (Frost et al. 2020) (Table S8). Interestingly, the modern strains displaying highest sequence identity to the partial Bolivian emm gene and that share a similar configuration of the mga regulon mostly come from the Americas or Oceania (Table S8 and S9, Figure S3).

Further analysis of the fibronectin-binding, collagen-binding, T antigen (FCT) region revealed that the ancient Bolivian *S. pyogenes* strain harbors a complete *Rof* regulon on contig 1, including the transcriptional regulator *rofA*, the fibronectin-binding protein *prtF1*, the backbone pilin *pilB9* (allele 15), the linker pilin *pilL9*, two pilus biosynthesis enzymes (sortases), and a putative LPxTG-anchored surface protein (*fctZ*) (Figure 2b). Based on the chromosomal organization of the FCT genes, the absence of an adhesin pilin gene (*pilA*), and the presence of *pilB9*, this strain is classified as carrying an FCT-9 region typical of group A streptococci (GAS) (Bessen et al. 2024).”

Additionally, we report the position of the two prophages on the genomic map and removed the heatmap of the MASH distances.

4. Where are the two intact prophages located? Please use the McShan et al nomenclature for this. When reporting the prophage elements in the text, please add which virulence genes are carried by which phage (and also in figure 3). Can the authors confirm whether the *ssa* gene (mentioned on page 8) was associated with a prophage (in line with modern genomes)?

Due to assembly fragmentation, we can't definitively say where the prophages are located. The prophage contigs are almost entirely prophage, so we're unable to infer their genomic context.

The only virulence gene we predicted in the prophage regions was VapE (described by Ji et al., 2016, Molecular Medicine Reports and later associated widely with prophages by Javan et al., 2019, Nature Communications).

Ssa was not found in this genome. *Ssa* was a false annotation due to 70% threshold of the VFDB, this is now corrected in the new version, and we confirm the absence of the super antigens *speA*, *speC*, and *sdn1* which are associated with modern phages, but we don't find them in the ancient genome.

5. *S. pyogenes* prophage typically are ordered by the integrase gene. In figure 3, it appears as though the assembly is disordered and the genetic makeup of the two prophage are very similar across several phage modules. This figure appears to be a pairwise alignment, yet this is not clear with the shading applied for the coloured phage modules. Suggest that the authors revise this figure to make the pairwise alignment more clear. The authors infer in their results that these assignments may be ambiguous. Can the assembly graphs be provided as supplementary to support their inferences. This confusion is reiterating in the final results paragraph, where the authors suggest degradation in these prophage and as such, may not represent “represent complete genomes with all necessary machinery for excision and replication“ as inferred in the preceding paragraphs.

We thank the reviewer for this interesting discussion. Figure 3 (new Figure S4a) showed one of the prophages (Bolivia Prophage 1). It is represented twice in the figure to allow for the different configurations of smaller prophage contigs that were not fully resolved in the assembly graph (shown in the tail and lysis sections on the right-hand side of the figure). These two configurations of the prophage are shown alongside a set of representative prophages taken from similar strains, to provide genomic context.

We're not entirely sure what is meant by “the assembly is disordered”. We've oriented all prophages in this figure in the same orientation (integrase to the left). The rationale to include a small section of host sequences downstream of this, was to show the genomic context (where possible, as many prophage contigs ended at the boundary of the prophage due to assembly fragmentation that is known to commonly occur at the termini of prophages).

The choice was made to illustrate conservation of blocks of phage proteins, rather than whole genome nucleotide alignments. This is due to phages typically being more diverse than their bacterial hosts, and their similarity is often described at the protein (AA) level rather than nucleotide. We believe this figure clearly illustrates the blocks of functional genes which are/aren't shared

between similar prophages and illustrates the mosaic architecture of this group of prophages, as has been noted previously in large scale studies of *Streptococcus* prophages (e.g. Javan *et al.*, 2019, Nature Communications).

6. Could the authors provide some comparison on the overall gene functional categories in the ancient genome relative to modern genomes? Are there any genes in the ancient genome that is NOT found in modern genomes (pan-genome analyses)? Was the number and type of gene function categories the same? Eg metabolic genes. eggNOG or equivalent would enable this functional categorisation of the strain relative to modern references. A reader may find this comparison interesting considering that diet changes that have occurred in the human host over time (or as the authors suggest may in fact be multi-host).

Using the same set of genomes (The HC245 clusters of the EnteroBase), we used for core-genome phylogenetic analysis, for further functional analysis using KEGG modules. As described in the methods, we used kofamscan to assign the genes to KEGG proteins using the threshold defined for each K0 protein. Then we mapped the K0 to the KEGG modules and used the gene counts per module for comparing different species.

As a results, we could not identify specific genes in the ancient *S. pyogenes* strain compared with other modern strains or the outgroup species. The full list of kegg-modules is shown in Tables S14, S15, and S16 as well as Figure 3b and Figure S5.

We additionally confirmed the previously reported differences between *S. pyogenes* and *S. dysgalactiae*, for example the M00018 of the threonine biosynthesis and the module M00854 of glycogen biosynthesis, glucose-1P => glycogen/starch are present in the *S. dysgalactiae* and missing from the *S. pyogenes*. While the module M00159 of the V/A-type ATPase, prokaryotes are more abundant in the *S. pyogenes* and missing in the *S. dysgalactiae*.

We amended the text with these differences as follows:

*“To further support the phylogenetic placement of both ancient strains, we aimed to identify species-specific KEGG modules using pangenome analysis. The final set of KEGG modules clearly indicate metabolic differences between the three species. Either complete KEGG modules or certain genes within modules are present in one species but not the other (Figure 3b and S5, Table S15, S16). As previously reported by Xie and colleagues (Xie et al. 2024) in their pangenome comparison of modern *S. pyogenes* and SDSE strains, we could confirm in our dataset the absence of the modules encoding glycogen biosynthesis (M00854) and threonine biosynthesis (M00018) in both the ancient Bolivian and modern *S. pyogenes* strains. Furthermore, unique to the ancient and modern *S. pyogenes* strains were the presence of multiple genes encoding V/A-type ATPases (M00159). There are additional differences in the metabolic gene set between the three *Streptococcus* species, such as unique presence of possible *cydAB* genes that encode subunits of the Cytochrome *bd* ubiquinol oxidase in SDSE, *S. canis* and the Gorilla strains. However, we do not see any indications for genes that occur only in the ancient Bolivian or Gorilla strain. In summary, the phylogenetic placement of the ancient Bolivian strain basal to the modern *S. pyogenes* and the Gorilla strain into the SDSE diversity is highly supported on pangenome level by the presence of species-specific genes in both ancient genomes.”*

7. Generally in ancient genome studies, the presence of an ancient phylogenetic root facilitate temporal style phylogenetic analysis. Perhaps the authors would consider Bayesian analysis to improve estimates regarding ancestral evolution of *S. pyogenes* (core genome)? Eg. Within the “HC10 clusters” (unclear to this reviewer what this cluster designation refers to) section where “evolutionary placement” phylogenetic inferences were undertaken.

We added the following text to the methods section to explain the HC cluster: *“To perform a comprehensive phylogenetic and comparative analysis, we utilized EnteroBase (<https://enterobase.warwick.ac.uk/>), a platform for exploring genomic epidemiology (Dyer et al. 2024).*

Specifically, we genomes of *Streptococcus pyogenes* and other two closely related species, i.e., *Streptococcus dysgalactiae* and *Streptococcus canis*. In EnteroBase, Hierarchical Clustering (HC) groups bacterial isolates based on core-genome multi-locus sequence typing (cgMLST) allelic profiles, with cluster levels (e.g., HC5, HC10, HC20, HC245) defined by the maximum number of allowable allelic differences, lower thresholds representing tighter, more recent genetic relationships. These clusters facilitate outbreak detection, epidemiological tracking, and population structure analysis by organizing isolates into hierarchically nested groups of increasing genetic diversity (Zhou et al. 2021). We opted to use the HC245 after exclusion of genomes with missing origin and isolation date in the metadata, which resulted in retrieving 149 genomes for *S. pyogenes*, 10 *S. dysgalactiae*, and 6 genomes for *S. canis*. A full list of genomes with metadata is available in Table SX. These genomes were used further for phylogenetic placement, pangenome analysis, and functional comparison.”

In the new version of the manuscript, we used the HC245 instead of the HC10 to reduce the number of genomes to facilitate the Bayesian phylogenetic analysis. We additionally included two outgroups (*S. canis* and *S. dysgalactiae*) + an ancient *S. dysgalactiae* genome we assembled from one of the gorilla samples (200-year-old).

We amended the results with the following text: “*Phylogenetic analysis shows that the ancient Bolivian S. pyogenes strain occupies a basal position and lies outside the diversity of known modern lineages. Despite this deep divergence, the strain possesses many genetic features associated with virulence in contemporary S. pyogenes, indicating that key pathogenic traits were already established early in the species’ evolutionary history. The Bolivian strain may therefore represent either an extinct lineage or a genomically unsampled clade that no longer circulates in present-day populations. Molecular dating suggests that the split between this ancient Bolivian strain and all other sampled S. pyogenes lineages occurred approximately 10,000 years ago. Archaeological evidence and excavations in the Bolivian highlands have revealed an extensive number of Archaic Period occupations. While radiometric dates are currently limited, the available evidence indicates a substantial human presence in the region starting as early as roughly 13,000 cal BP (47). As humans first moved into the Andes, they underwent a sophisticated process of adapting both their physical biology as well as their social customs (48). Both these processes and the strategic evolutionary responses, like potential pathogen adaptations and interactions were developed to challenge and thrive in such a unique landscape. The divergence between modern and ancient S. pyogenes strains post-dates the widely accepted migration of humans into the Americas via the Bering Strait approximately 22,000 years ago (49). Genomic evidence shows multiple gene flow events and regional population turnover across the Americas during the Holocene (<10 kya) (50,51), suggesting dynamic intra-continental population structure that may have influenced the pathogen’s spread. While this timing is compatible with scenarios involving zoonotic transmission from an as-yet unidentified host or the introduction of an ancestral strain during post-glacial migrations of Siberian populations into the Americas within the past 10,000 years (52), these possibilities remain speculative. The basal phylogenetic placement of the ancient Bolivian strain and reconstructions of the most recent common ancestor may be consistent with an American origin of the pathogen; however, these observations alone are insufficient to support such a conclusion. Basal positioning can also result from the extinction of lineages in other regions or from incomplete sampling. Consequently, additional global representatives of ancient and modern S. pyogenes genomes will be required to more rigorously evaluate the robustness of the inferred divergence dates and phylogenetic relationships. By expanding our search for S. pyogenes to other publicly available ancient DNA datasets, we detected the pathogen in humans as early as 4,000 years ago in Europe and Africa. In addition, we observed traces of SDSE DNA, the closest phylogenetic relative of S. pyogenes, in gorillas around 200 years ago, suggesting a complex evolutionary history of both pathogens. The presence of S. pyogenes across geographic regions and time periods raises the possibility that it was carried over by human populations during their migrations, contributing to its global distribution. Our current Bayesian phylogenetic analyses indicate that the majority of modern S. pyogenes lineages diversified much more recently, largely within the past ~5,500 years of human history. This recent diversification likely reflects major changes in human population structure during the late Holocene in different parts of the world”*

8. In the discussion, the authors extrapolate their findings to infer invasive potential “These findings also suggest that it would belong to an invasive group A Streptococcus (GAS) lineage capable of causing severe infections in ancient human populations.” This is a loose inference, and should be tempered.

We agree with the reviewer particularly that we didn't find the *speA*, *speC* and *sdn1* genes which would make it invasive. We removed this claim from the text and confined the virulence discussion to the well documented and annotated genes.

Minor comments:

Lowercase emm when referring to the gene (even at the start of a sentence). Eg. Emm137 should be emm137

Corrected throughout the text

Please italicise “Streptococcus” throughout.

Corrected

Please double check references. There appears to be some duplication (eg 39 and 5).

Checked and corrected

Reviewer #3 (Remarks to the Author):

This is an extremely interesting data set. My primary criticism (points #1-#5) is that the sequence data has not been analyzed to sufficient depth. There are some key questions that remain unanswered - however they could be readily addressed, and such an analysis is likely to elevate the impact and value of the study. A second concern lies in speculations on data interpretation (points #6-#9).

Overall, I find some of the summations of the *S. pyogenes* and related literature to be simplified and/or out of context (and probably unnecessary). The manuscript could likely be strengthened by a more focused approach that includes a more thorough evolutionary analysis of this important genome sequence. Specific comments are listed below:

1. One key piece of missing analysis is an in-depth comparison of the Bolivian isolate genome to *Streptococcus equisimilis* subspecies *dysgalactiae* (SDSE) – SDSE are the closest known genetic relatives of *S. pyogenes*. SDSE and *S. pyogenes* display an extensive history of horizontal gene exchange and share a very large number of virulence genes (*emm*, FCT-9 pili, streptolysins, streptokinase, many others), core genes (including MLST overlap) and mobile genetic elements (phage genes, R-genes). For example, please see Xie et al (Nature Comm, 2024), among several other publications on *S. pyogenes*-SDSE gene flow. A deeper analysis of the Bolivian isolate versus closely related groups C and G strep genomes seems warranted. *Strep. agalactiae* was chosen for comparison as a related species, however, it is more genetically distant.

Thanks for the reviewer for the insightful comment. Based on initial screening we carried out using the Type (Strain) Genome Server (<https://tygs.dsmz.de/>), the closest species based on the complete genome sequence is *S. dysgalactiae*

and the closest species based on 16S rRNA is *S. canis*.

We include the two species as outgroup in the new phylogenetic analysis and in the functional comparisons.

In the new Figure 2a, we report the gene clusters/operon of the main regulatory/virulence genes which can help to contextualise the ancient strain. Namely, we report the gene clusters of FCT regions, mga regulon, Streptolysin S (sag operon), and hasABC genes.

In the results section, we described the arrangements of the *mga* and FCT regions in a way that can textualize them. The new text added is as follows:

“To systematically and functionally assign the ancient S. pyogenes strain we applied multi-locus sequence (MLST), emm, and pilin typing schemes that are commonly used to classify modern S. pyogenes strains (Bessen et al. 2018b; Bessen et al. 2022). MLST typing of seven housekeeping genes assigned the ancient strain to a distinct sequence profile, that has no direct modern equivalents and possibly represents an ancestral lineage of contemporary strains (Table S6). The most similar modern S. pyogenes isolates come from remote aboriginal island communities in tropical Australia sharing three out seven MLST alleles with the ancient strain (McGregor et al. 2004). Additional S. pyogenes typing schemes target the emm gene encoding the surface-exposed streptococcal M protein, and pilin adhesin and backbone genes (Bessen et al. 2024). Both M proteins and pili are key virulence factors and targets of host protective immunity (Walker et al. 2014). The emm gene is located in the mga regulon with genes encoding additional M-like proteins such as mrp and enn, and scpA encoding a C5a peptidase (Hondorp & McIver 2007). In the assembled Bolivian genome, the mga regulon genes are distributed to two different contigs, contig 17 that carries the transcriptional activator gene mga and a partial emm gene, and contig 13 that encodes the emm-like gene enn and the C5a peptidase gene scpA (Figure 2b). Considering the size range of 220 to 513 amino acids in modern M protein sequences (Frost et al. 2020), the emm gene of the ancient Bolivian strain is nearly complete (1161 bp, 387 aa in length) lacking only the 3' gene part that encodes the D-repeat region of the M protein (Figure S3). This gene part could not be assembled with the current available data situation and assembly strategy. Nevertheless, emm typing targeting the 5' region of the gene was possible and assigned the ancient S. pyogenes strain closest (79% nucleotide identity) to the modern emm type 46 (see Figure S4, Table S7). However, based on the currently existing typing criteria, the ancient strain should be assigned to a unique emm type, having less than 92% nucleotide identity over the first 90 bases encoding the mature M protein to any other emm type (Bessen et al. 2018a). Next, we extended our strain typing to the emm pattern analysis that groups emm types into functional patterns (A–C, D, E) that correlate with tissue tropism, ecological niche, and pathogenic behaviour (Bessen et al. 2022). Based on the chromosomal arrangements of emm and emm-like genes (mga-emm-enn-scpA, Figure 2b) and their subfamily (SF) forms (enn SF-1), the ancient Bolivian strain can be associated with the emm pattern group A-C, being most likely a S. pyogenes emm pattern B strain. The analysis of a global collection of Group A Streptococcus revealed that majority of modern strains belong to emm pattern groups D and E, whereas emm pattern B or C strains are nowadays only rarely observed (Frost et al. 2020) (Table S8). Interestingly, the modern strains displaying highest sequence identity to the partial Bolivian emm gene and that share a similar configuration of the mga regulon mostly come from the Americas or Oceania (Table S8 and S9, Figure S3).

Further analysis of the fibronectin-binding, collagen-binding, T antigen (FCT) region revealed that the ancient Bolivian S. pyogenes strain harbors a complete Rof regulon on contig 1, including the transcriptional regulator rofA, the fibronectin-binding protein prtF1, the backbone pilin pilB9 (allele 15), the linker pilin pilL9, two pilus biosynthesis enzymes (sortases), and a putative LPxTG-anchored surface protein (fctZ) (Figure 2b). Based on the chromosomal organization of the FCT genes, the absence of an adhesin pilin gene (pilA), and the presence of pilB9, this strain is classified as carrying an FCT-9 region typical of group A streptococci (GAS) (Bessen et al. 2024).”

Using the same set of genomes (The HC245 clusters of the Enterobase), we used for core-genome phylogenetic analysis, for further functional analysis using KEGG modules. As described in the methods, we used kofamscan to assign the genes to KEGG proteins using the threshold defined for each K0 protein. Then we mapped the K0 to the KEGG modules and used the gene counts per module for comparing different species.

As a results, we could not identify specific genes in the ancient *S. pyogenes* strain compared with other modern strains or the outgroup species. The full list of KEGG-modules is shown in Tables S14, S15, and S16 as well as Figure 3b and Figure S5. We additionally confirmed the previously

reported differences between *S. pyogenes* and *S. dysgalactiae*, for example the M00018 of the threonine biosynthesis and the module M00854 of glycogen biosynthesis, glucose-1P => glycogen/starch are present in the *S. dysgalactiae* and missing from the *S. pyogenes*. While the module M00159 of the V/A-type ATPase, prokaryotes are more abundant in the *S. pyogenes* and missing in the *S. dysgalactiae*.

We amended the text with these differences as follows:

“To further support the phylogenetic placement of both ancient strains, we aimed to identify species-specific KEGG modules using pangenome analysis. The final set of KEGG modules clearly indicate metabolic differences between the three species. Either complete KEGG modules or certain genes within modules are present in one species but not the other (Figure 3b and S5, Table S15, S16). As previously reported by Xie and colleagues (Xie et al. 2024) in their pangenome comparison of modern S. pyogenes and SDSE strains, we could confirm in our dataset the absence of the modules encoding glycogen biosynthesis (M00854) and threonine biosynthesis (M00018) in both the ancient Bolivian and modern S. pyogenes strains. Furthermore, unique to the ancient and modern S. pyogenes strains were the presence of multiple genes encoding V/A-type ATPases (M00159). There are additional differences in the metabolic gene set between the three Streptococcus species, such as unique presence of possible cydAB genes that encode subunits of the Cytochrome bd ubiquinol oxidase in SDSE, S. canis and the Gorilla strains. However, we do not see any indications for genes that occur only in the ancient Bolivian or Gorilla strain. In summary, the phylogenetic placement of the ancient Bolivian strain basal to the modern S. pyogenes and the Gorilla strain into the SDSE diversity is highly supported on pangenome level by the presence of species-specific genes in both ancient genomes.”

2. A second key piece of missing analysis: Most *S. pyogenes* strains have an emm gene plus 1 or 2 emm-like (emmL) genes; thus, one cannot necessarily assign emm-type based on a sequence hit alone. Evidence should be provided showing that the so-called ‘emm137’ gene corresponds to the correct genome map position for the type-specific emm locus - and not the downstream (paralogous) emmL/enn locus. Most emm genes have a history of horizontal transfer and recombination, and most emm types have been recovered from multiple divergent genetic backgrounds. The (rare) emm137 Fiji strain cited in reference 29 has a distinct FCT-region (not FCT-9) and therefore, does not appear share a recent/direct common ancestor with the Bolivian strain.

3. Related to #2: An in-depth and granular Blast analysis that includes the entirety of the emm-region – spanning from the Mga locus to the C5a peptidase locus (whether assembled or in fragments) – might address the emm/emmL ‘137’ issue, as well as provide key evolutionary insights since the emm-region of the genome is subject to strong diversifying selection. It may be insightful to know which sub-portions of the emm-region and emm/emmL gene product functional domains display high homology to contemporary *S. pyogenes* genes, and which parts are unique and non-extant.

Addressed in the previous comment

4. A map of the FCT region genes might also be a nice addition. It is stated that the Bolivian isolate displays the FCT-9 region form. Is the pilB gene highly homologous to any contemporary alleles? How about other FCT-9 region genes? Several FCT-9 loci of *S. pyogenes* are shared with group C and G streptococci.

The FCT-9 region is displayed in Figure 2a and the homology results are shown in Table S10.

The homology of the pilB9 and pilL9 were highly similar to modern strains however the prtF1 highest similarity was 88.26%. The overall structure of the cluster is described in the text as shown in the previous comment.

category	cluster/gene	contig	start	end	strand	coverage	identity	accession	gene	ID
Pilus structure	T-Protein FCT-Region	contig_1	9865	11358	-	100	98.79	MFF1072351.1	transcriptional regulator RofA	RofA
		contig_1	11626	13638	+	100	88.26	HEP1412083.1	fibronectin-binding protein PrtF1/SfbI	PrtF1/SfbI
		contig_1	14867	16582	+	100	100	WP_136116429.1	SpaH/BbpB family LPXTG-anchored major pilin	PilB9
		contig_1	16664	17596	+	100	99.68	WP_136116430.1	SpaA isozeptide-forming pilin-related protein	PilL9
		contig_1	17598	18461	+	100	99.65	HEP2959403.1	class C sortase	srtC
		contig_1	18463	19320	+	100	100	WP_111677010.1	class C sortase	srtC
		contig_1	19382	24538	+	100	99.65	VHF67001.1	fibronectin-binding protein	sdrE
		contig_1	24715	25380	-	100	95.48	WP_326610848.1	hypothetical protein [Streptococcus pyogenes]	
		contig_1	25729	27135	+	100	99.79	AAT86298.1	short-chain fatty acid transporter	

5. One of the benefits of a dated ancient genome is that the data can be used to refine the ‘molecular clock’ and mutation rates. Although often such analyses yield soft conclusions, a thoughtful attempt here might be worthwhile. A deeper analysis of select core genes, for their sequence divergence with modern-day strains, may also provide interesting new evolutionary insights. Additional data supporting the novelty of the ancient isolate can help to minimize any lingering doubts that it is a modern-day contaminant.

We agree with the reviewer. In the revised phylogenetic analyses, after including the outgroups, the ancient Bolivian *Streptococcus pyogenes* genome (Bolivia 2730) falls in a basal position relative to the diversity of modern *S. pyogenes*. In both maximum-likelihood and Bayesian molecular clock frameworks, the ancient genome branches off prior to the radiation of contemporary strains, consistent with an early-diverging lineage within the species.

In the new figure 3, we described the new positioning and the estimated divergent time of the ancient *S. pyogenes* from the modern strains and amended the text as follows: ““*To further support the phylogenetic placement of both ancient strains, we aimed to identify species-specific KEGG modules using pangenome analysis. The final set of KEGG modules clearly indicate metabolic differences between the three species. Either complete KEGG modules or certain genes within modules are present in one species but not the other (Figure 3b and S5, Table S15, S16). As previously reported by Xie and colleagues (Xie et al. 2024) in their pangenome comparison of modern S. pyogenes and SDSE strains, we could confirm in our dataset the absence of the modules encoding glycogen biosynthesis (M00854) and threonine biosynthesis (M00018) in both the ancient Bolivian and modern S. pyogenes strains. Furthermore, unique to the ancient and modern S. pyogenes strains were the presence of multiple genes encoding V/A-type ATPases (M00159). There are additional differences in the metabolic gene set between the three Streptococcus species, such as unique presence of possible cydAB genes that encode subunits of the Cytochrome bd ubiquinol oxidase in SDSE, S. canis and the Gorilla strains. However, we do not see any indications for genes that occur only in the ancient Bolivian or Gorilla strain. In summary, the phylogenetic placement of the ancient Bolivian strain basal to the modern S. pyogenes and the Gorilla strain into the SDSE diversity is highly supported on pangenome level by the presence of species-specific genes in both ancient genomes.*”

6. Descriptive annotations derived from bioinformatics software do not by themselves equate to biological function. There are several instances in the manuscript where the authors delve into commentaries on annotated genes as virulence- or resistance-associated - yet those ascribed gene functions in this species (*S. pyogenes*) are not necessarily experimentally proven phenotypes. Unless biological functions are supported by (accurate) citations, the discussions seem way too speculative; this concern extends to the Abstract.

We confined the virulence discussion to the well-documented gene clusters as indicated by the reviewers.

7. What is meant by “invasive” lineage? Most strains/emm-types of *S. pyogenes* can (more or less) cause invasive disease.

We agree with the reviewer, therefore, in the new version we didn't use the term “invasive” to describe ancient strain.

8. The commentary on a systemic infection, with a possible mixed infection, seems weak and not well-connected to current clinical knowledge. The ‘co-isolates’ *Clostridium tetani* and *botulinum* (and *Morganella*) are primarily environmental (soil; and the clostridia form endospores, perhaps preserving their DNA); it is extremely rare for these species to be present in the bloodstream (unlike *S. pyogenes*). Perhaps the person had sepsis due to *S. pyogenes* or perhaps they had a polymicrobial tooth abscess (although a mixed infection would more likely include anaerobes from the oral microbiota), or maybe the cause of death was non-infectious – one can only speculate.

We confined the discussion now on the *S. pyogenes* and toned down the mixed infection possibility

9. The discussion on the population structure of *S. pyogenes* seems a bit out of place and unclear.

In the new phylogenetic analysis, the Bolivian strain falls basal to all modern diversity of *S. pyogenes*. The molecular clocking suggests split time around 10270 years BP. Please refer the comment #5

10. There are a few reports of natural recovery of *S. pyogenes* from non-human primates, although there is no strong evidence that this bacterium is a pathogen of consequence (thereby ‘human-specific’). The authors may wish to consider their discussion within this context.

11. The references lack journal titles. Some citations do not provide the evidentiary support that the authors claim.

Revised and corrected

12. Line numbers would be helpful.

Added

Reviewer #4 (Remarks to the Author):

In this manuscript, Valverde and colleagues present the ancient DNA analysis of a tooth sample from a pre-Columbian era mummy from Bolivia. The highlight of the study is the recovery and characterization of a near-complete genome of a pathogen, *Streptococcus pyogenes*.

The paper is generally well-written. The analyses related to the phylogeny and genomic architecture of the *S. pyogenes* strain are well done and interesting to read. However, the authors need to do a better job of convincing the readers regarding the ancient origin of the pathogens presented in this paper. The following comments need to be addressed before I can recommend the manuscript for publication:

The authors state “Further microbial profiling using MetaPhlAn4 displayed unusual high abundances of different human pathogenic bacterial species - *Clostridium tetani* (37.5%), *Morganella morganii* (29.5%), *Streptococcus pyogenes* (11.6%), *Clostridium botulinum* (6.7%), *Asaccharospora irregularis* (6.4%), and *Paeniclostridium sordellii*“. Typically, when one is analyzing ancient pathogen DNA from a tooth sample, the pathogen’s DNA is very low in abundance as compared to that from oral microbes. The profile given here strikes me as very unusual. Given the high abundance of these six taxa in the sample, one might hypothesize that these are contaminants that have leached into the tooth from the burial environment. This possibility needs to be discussed and the authors need to make a better case for why they think these are ancient taxa, endogenous to the tooth sample, and why their presence in this sample indicates that the individual suffered from an infection by these pathogens. For e.g. *Clostridium tetani* and *C. botulinum* can both be found in the soil. *Asaccharospora irregularis* was formerly considered to be a *Clostridium*, so it is likely found in the soil too, but I can find no reference of it being a human pathogen.

We are not highlighting the mixed infection possibility anymore, although we highlight the incidence of the other species. The high abundances of the pathogens particularly *S. pyogenes*, indicates the exceptional preservation of DNA retrieved from this mummy.

Additionally, the DNA was extracted from the tooth pulp after disinfecting the tooth external surface. Also, it is worth noting that the damage profiles were low compared to other ancient samples from the same age.

As a further proof of the preservation, we show in Figure S2, the damage profiles of the reads mapped to the assembled genome before and after applying PMD tools filtering which excludes undamaged reads. What we observe here is that we have a high peak of reads at 151 nt (max read length we get from the sequencer, and they are relatively long compared to ancient DNA). This peak persisted as we increase the strictness of the PMD tools filters as shown in the figure below (Figure S2).

From Table 1, it seems the *S. pyogenes* DNA shows 1.2% and 1.7% damage at the 5' and 3' ends, respectively, whereas the human DNA from the same tooth shows 3.7% and 5.1% damage. The authors need to discuss this discrepancy between the amount of damage for the host and pathogen DNA.

We have observed a similar discrepancy even when comparing human DNA from different tissues of the same mummy. For example, from the Barfusser mummy of Anna Catharina Bischoff, we observed higher DNA damage levels in the tooth DNA compared to the human DNA from soft tissues (Sarhan et al. 2023; <https://link.springer.com/article/10.1186/s12915-022-01509-7>). In that case, the reason was attributed to different concentrations of mercury in different body part which could reduce the hydrolytic damages and other microbial activities.

The argument for mixed infection seems weak to me, especially since I find it less likely that the *Clostridium* species are endogenous to the tooth sample. In any case, the authors will need to give damage patterns for the other pathogens in order to support an argument for mixed infection.

We are not highlighting the mixed infection anymore, we only discuss the *S. pyogenes* and we referred to the other microbes as opportunistic pathogens.

It is disappointing that there is no discussion of the burial context for this individual. Furthermore, was any paleopathological analysis done that suggests the presence of any particular disease(s)? Also, please specify which tooth was sampled.

-We have added the description of the funerary and burial contexts in a new section titled: Burial context and palaeopathological overview. The text read as follows:

"The mummified remains housed in the National Museum of Archaeology were recovered from the *Chullpas*, or funerary towers, spread across the Bolivian Altiplano. During the Late Intermediate Period (1100 – 1450 AD), these individuals were interred within funerary bundles and placed inside these structures. While there are some accounts of anthropogenic mummification practices, the majority of the remains found in this region likely underwent a process of natural mummification over time."

-We have reviewed the initial bioarchaeological profiling done in individual ID 2730 to complete the background information. We provide now the additional information in the text as follows:

"The palaeopathological examination suggests that this individual underwent intentional cranial modification. Dental assessment reveals dental chipping and localized periapical infections involving both mandibular first molars (M₁). A linear fracture is observed on the inferior portion of the occipital bone, extending posteriorly from the foramen magnum along the sagittal plane. Furthermore, post-mortem fractures of the cervical vertebrae, specifically involving the axis, show evidence of mechanical tension consistent with the deliberate removal of the cranium after death"

The DNA extraction and sequencing section need more details. Please briefly describe the extraction and library preparation protocol used.

We have added details about the DNA sampling process and the DNA extraction protocol. The text reads as follows:

"DNA extraction was conducted following the protocol developed by Fellows Yates et al. (2021), specifically optimized for the recovery of ancient pathogens. The procedure involved the transverse sectioning of the dental crown, followed by the mechanical drilling of the pulp chamber, which serves as a reservoir for endogenous pathogen DNA. This area is targeted because the highly vascularized pulp tissue historically trapped circulating pathogens, preserving their genetic material within the tooth structure"

We have added the following text for the library preparation: "The method focuses on "double-indexing" to maximize sample identification during high-throughput sequencing. It involves blunt-end repair, adapter ligation, and a specialized indexing PCR to create libraries ready for Illumina platforms".

1.

Bessen, D.E., Beall, B.W. & Davies, M.R. (2022). Molecular basis of serotyping and the underlying genetic organization of *Streptococcus pyogenes*. *Streptococcus pyogenes: Basic Biology to Clinical Manifestations*. 2nd edition.

2.

Bessen, D.E., Beall, B.W., Hayes, A., Huang, W., DiChiara, J.M., Velusamy, S. et al. (2024). Recombinational exchange of M-fibril and T-pilus genes generates extensive cell surface diversity in the global group A *Streptococcus* population. *J MBio*, 15, e00693-00624.

3.

- Bessen, D.E., Smeesters, P.R. & Beall, B.W. (2018a). Molecular epidemiology, ecology, and evolution of group A streptococci. *Microbiology spectrum*, 6, 10.1128/microbiolspec.cpp1123-0009-2018.
- 4.
- Bessen, D.E., Smeesters, P.R. & Beall, B.W. (2018b). Molecular Epidemiology, Ecology, and Evolution of Group A Streptococci. *Microbiology spectrum*, 6.
- 5.
- Buckley, S.J., Timms, P., Davies, M.R. & McMillan, D.J. (2018). In silico characterisation of the two-component system regulators of *Streptococcus pyogenes*. *PLoS one*, 13, e0199163.
- 6.
- D'Costa, V.M., King, C.E., Kalan, L., Morar, M., Sung, W.W., Schwarz, C. *et al.* (2011). Antibiotic resistance is ancient. *477*, 457-461.
- 7.
- Dyer, Nigel P., Päuker, B., Baxter, L., Gupta, A., Bunk, B., Overmann, J. *et al.* (2024). EnteroBase in 2025: exploring the genomic epidemiology of bacterial pathogens. *Nucleic Acids Research*, 53, D757-D762.
- 8.
- Frost, H.R., Davies, M.R., Delforge, V., Lakhroufi, D., Sanderson-Smith, M., Srinivasan, V. *et al.* (2020). Analysis of global collection of group A *Streptococcus* genomes reveals that the majority encode a trio of M and M-like proteins. *Msphere*, 5, 10.1128/msphere.00806-00819.
- 9.
- Hondorp, E.R. & Mclver, K.S. (2007). The Mga virulence regulon: infection where the grass is greener. *Molecular microbiology*, 66, 1056-1065.
- 10.
- McGregor, K.F., Bilek, N., Bennett, A., Kalia, A., Beall, B., Carapetis, J.R. *et al.* (2004). Group A streptococci from a remote community have novel multilocus genotypes but share emm types and housekeeping alleles with isolates from worldwide sources. *189*, 717-723.
- 11.
- Walker, M.J., Barnett, T.C., McArthur, J.D., Cole, J.N., Gillen, C.M., Henningham, A. *et al.* (2014). Disease manifestations and pathogenic mechanisms of group A *Streptococcus*. *27*, 264-301.
- 12.
- Xie, O., Morris, J.M., Hayes, A.J., Towers, R.J., Jespersen, M.G., Lees, J.A. *et al.* (2024). Inter-species gene flow drives ongoing evolution of *Streptococcus pyogenes* and *Streptococcus dysgalactiae* subsp. *equisimilis*. *Nature communications*, 15, 2286.
- 13.
- Zhou, Z., Charlesworth, J. & Achtman, M. (2021). HierCC: a multi-level clustering scheme for population assignments based on core genome MLST. *Bioinformatics*, 37, 3645-3646.

Point-by-point response to the reviewers' comments

Second round of review

Our responses in blue

REVIEWERS' COMMENTS

Reviewer #1 (Remarks to the Author):

I thank the authors for their careful and thorough revision. My concerns have been adequately addressed, and the manuscript has been substantially strengthened. I have no further comments and support publication in its current form.

We thank the reviewer for this positive assessment

Reviewer #2 (Remarks to the Author):

The authors have made substantial alternations to the manuscript including new analyses which address the major concerns/queries. The revised version has substantially improved and the authors should be commended for their diligence. The supplementary material is important for validation.

We thank the reviewer for this positive assessment

While the raw data looks to have been deposited, it is strongly recommended that draft assemblies generated be made public available.

Sequencing data and the assembled genome are available at the European Nucleotide Archive (ENA) under ENA: PRJEB91735.PRJEB91735. The sequencing reads are available at the Sequence Read Archive under accession ERR15308372 ERR15308372 and the assembly under accession number GCA_982145515.1.GCA_982145515.1. Source data are provided with this paper.

Reviewer #3 (Remarks to the Author):

The extensive additional analyses and thoughtful responses to reviewer concerns are greatly appreciated. The revised manuscript seems much better focused and importantly, seems well-poised to have a high impact on the *S. pyogenes* field.

I have a few suggestions for further clarifications:

1. Lines 79-80. "Like *S. pyogenes*, SDSE commonly expresses Lancefield group C or G antigens....." Incorrect as written: *S. pyogenes* express Lancefield group A carbohydrate.

Thanks for pointing out this incorrect phrasing. We rewrote the sentence as follows:

"In contrast to S. pyogenes, that belongs Lancefield serogroup A, SDSE commonly expresses Lancefield group C or G antigens, but still causes a similar spectrum of human diseases ranging from pharyngitis to invasive soft tissue infections."

Also, if the authors know that their isolate has high homology to the group A carbohydrate biosynthesis genes (*gac* genes), it may be of added value to include that point in their analysis of other key genes (Results section)

We followed the reviewer's suggestion and subjected the twelve *gac* operon genes located on the contig 5 to BlastP homology analysis and included the results into the Supplementary Data 10 table. All twelve *gac* genes show high homology (>99%) to *S. pyogenes gac* genes.

We added the following sentence to the results section of the revised manuscript:

"The full set of genes (gacA-L) of the group A carbohydrate (GAC) gene cluster are present on contig 5 with high homology, i.e., > 99% proteins identity, to modern GAS strains (Supplementary Data 10)."

2. Line 83: Omit the "Thereby" – Exchange of many [virulence] genes is by mechanisms unknown (i.e., not yet proven for mobile genetic elements as the vehicle, except for [largely] pyrogenic exotoxins via transduction]). Also lines 85-87.

We omitted the "Thereby" and used the pyrogenic exotoxins as an example.

3. Line 253 (and related Discussion): animal infection models for pneumococcus (not *S. pyo*)? Ref. 35

Thank you for pointing this out. We changed the part in Line 253 accordingly and wrote: *"Despite the absence of superantigens encoded on these two prophages, one prophage harboured a virulence determinant, Virulence-Associated Protein E (VapE), that was previously associated with virulence in a mouse model of Streptococcus pneumoniae infection."*

The Discussion part has been removed.

4. Line 276-paragraph (and related Discussion): Is the *ImrP* gene product actually known to confer antibiotic resistance in this species (*S. pyo*)? Citation? Also, a functional 'mef[E]' is far more typical for other streptococcal species. The ideas in this section seem a bit overstated.

Thank you for pointing this out. We rephrased the last sentence in line 253 accordingly:

"Both genes were complete, with no evidence of frameshifts or premature stop codons, suggesting they were intact in the ancient strain. Their functional properties, however, remains to be determined."

In addition, we removed the following sentence in the discussion:

"The high sequence identity and broad coverage observed indicate that these genes were likely functional at the time, supporting the hypothesis that resistance traits were already circulating in microbial communities before the antibiotic era"

5. Lines 396-399: It is not all that uncommon for modern strains to lack phage-associated virulence (superantigen) genes. So perhaps this idea is not quite correct as presented.

We agree and removed the following sentence:

“Their absence in the ancient genome supports the view that such superantigens represent later-acquired, mobile genetic elements rather than ancestral core virulence factors, highlighting the role of horizontal gene transfer in increasing the pathogenic potential of modern S. pyogenes strains.”

6. Line 400: “S. pyogenes is predominantly a clonal pathogen” (ref. 43 cited). Authors should probably clarify or soften. Global strains of S. pyogenes taken as a group are nearly as ‘recombinogenic’ as pneumococci via numerous measures. There exist some strongly ‘clonal’ lineages within S. pyogenes (e.g., M1T1) and many lineages/clusters (phylogroups) that are more diverse

Thank you for pointing this out! We rephrased the sentence accordingly:

“S. pyogenes lineages can be grouped into related clusters (‘phylogroups’) defined by whole genome clustering methods. These phylogroups are not genetically static, as occasional recombination occurs via mobile genetic elements such as...”

Reviewer #4 (Remarks to the Author):

The manuscript by Valverde and colleagues is much improved following the previous round of revision. I am satisfied with their response to the concerns I had previously raised. Below I have noted some additional (mostly minor) concerns that need to be addressed.

Lines 319-324: The authors infer that the timing of the origin of S. pyogenes is the same as the timing of its split from S. dysgalactiae i.e. 137,264 years. I don’t quite agree with this inference; in my opinion, the timing of the MRCA of all S. pyogenes strains (ancient and modern), which is estimated to be 10,270 years, is what one should consider as the origin of this species. If the authors disagree with this, I encourage them to include a sentence stating why they think using the stem age (time at which S. pyogenes diverged from its sister taxon), as opposed to the crown age (time of MRCA for all S. pyogenes), would be appropriate to consider as the time at which this species originated.

We agree with the reviewer that our wording was unclear and that the crown is more correct for describing the age of S.pyogenes. Technically, the origin of S.pyogenes is anytime between the stem and the crown, with more confidence as we get closer to the Crown.

We rephrased the part accordingly:

“Time estimates of the full dataset under most fitting analytical parameters (see Methods) returned an age for the split of S. pyogenes and S. dysgalactiae spp. at 137,264 years (95% HPD: 124,428 to 149,847 cal years BP) (Figure 3a). The timing of the Most Recent Common Ancestor (MRCA) of all S. pyogenes strains (ancient Bolivian strain and modern known S. pyogenes) is estimated to be 10,270 years (95% HPD: 9314 to 11,218 cal years BP). The diversification of most modern S. pyogenes took place in the past 5,500 years of human history (Figure 3C).”

Line 532: Please specify which bwa algorithm was used – mem or aln? Were default parameters used? If using aln, were recommended aDNA parameters used (such as disabling the seed, changing mapping stringency to allow reads with damaged nucleotides to map)? If yes, please specify the parameter values.

We specified accordingly:

“For the analysis of the human DNA, we mapped the merged reads against the reference human genome (build hg19), using BWA-aln v0.7.17 59 (using the following parameters: -l 1000 -n 0.04 -o 1 -e -1), then filtered for minimum mapping quality of 25 using SAMtools v1.17 60.”

Line 533: With regards to Qualimap, do you mean to say coverage instead of mapping quality?

We rephrased accordingly:

“The mapping quality and coverage were then checked using Qualimap v2.2.2d 63.”

Line 536: It is unclear how the consensus sequence for the human DNA was called, since this cannot be done using either MapDamage2 or Haplogrep. Did you mean to say you rescaled the damaged sites using MapDamage2 and then called the consensus using schmutzi?

We rephrased accordingly:

“The damaged sites of aDNA reads in the BAM file were then rescaled using MapDamage 2 and used to call a consensus sequence using Schmutzi script “log2fasta”, for mitochondrial haplogroup assignment using Haplogrep3 v2.4.0 64.”

Line 631: Correct "EES" to "ESS". Also, specify what was considered as a “satisfactory” ESS value.

We rephrased accordingly:

“All MCMC were run for a minimum of 50 million generations until convergence. Convergence was inspected using Tracer by checking that all parameters had an ESS>200 and setting a burn-in that maximised ESS.”

Apart from the above minor concerns, the manuscript needs to be thoroughly proof-read to correct instances of awkward phrasing and lingering typos. The instances I could find are given below.

Line 30: Missing period at end of sentence. Add “the” before “European colonial expansion”.

Checked (new Abstract)

Line 37: Use “the” in place of “an”.

Checked (new Abstract)

Line 56: Add comma after “health”.

Checked and corrected

Lines 96, 124: Add “the” before “pre-Columbian”.

Checked and corrected

Line 104: Missing period after the “S” of Streptococcus.

Checked and corrected

Line 105: Rephrase to “... and calibrate the pathogen’s phylogeny...”

Checked and corrected

Line 109: Rephrase to “...tooth sample from a pre-Columbian...”

Checked and corrected

Line 129: Rephrase to “...environments with dry conditions which have been previously shown to be less prone to DNA degradation.”

Checked and corrected

Line 130: Remove space between MetaPhlAn and 4.

Checked and corrected

Line 138: Rephrase to “...to this species to understand...”

Checked and corrected

Line 140-142: Rephrase to “...which revealed the presence of *S. pyogenes* in human samples from different time periods (ranging from 4000 to 200 years BP) in Europe and Africa, as well as in gorilla samples from museum collections”

Checked and corrected

Line 144: I recommend using the word “samples” instead of the word “findings”.

Checked and corrected

Line 164: Rephrase to “The completeness of the assembled *S. pyogenes* genome was 99.98%..”

Checked and corrected

Table 1: Needs to be properly formatted with respect to gray and white cell colors.

Checked and corrected

Line 179: Redundant “typing” after MLST.

Checked and corrected

Line 195: Remove the word “situation”. I recommend using “currently available” instead of “current available”.

Checked and corrected

Line 197: Rephrase to “...*pyogenes* strain to the modern emm type 46 (79% nucleotide identity)”

Checked and corrected

Line 234: Rephrase to “In addition to the surface-associated M- and T-proteins, the ancient *S. pyogenes* strain holds...”

Checked and corrected

Line 252: Change to "...pathogenicity in animal infection..."

Checked and corrected

Line 255: Specify here what you mean by "our prophages". I recommend saying "however, the prophages in the ancient strain...".

Checked and corrected

Line 290: Rephrase to "to be completely absent in other modern GAS genomes as well".

Checked and corrected

Line 297, 311, 312, 314, and elsewhere: Do not capitalize the word "gorilla".

Checked and corrected

Line 317: Remove the word "ancestral". It is redundant in that sentence.

Checked and corrected

Line 322 – 324: Rephrase to "The split of the Bolivian strain from all other *S. pyogenes* (divergence of known *S. pyogenes*) was estimated to be 10,270 years ago (95% HPD: 9314 to 11,218 cal yr BP), and a diversification of most modern *S. pyogenes* in the past 5,500 years of human history."

We rephrased to:

"The timing of the Most Recent Common Ancestor (MRCA) of all S. pyogenes strains (ancient Bolivian strain and modern known S. pyogenes) is estimated to be 10,270 years (95% HPD: 9314 to 11,218 cal years BP). The diversification of most modern S. pyogenes took place in the past 5,500 years of human history (Figure 3C)."

Line 370: Missing period after the word "infections".

Checked and corrected

Line 476: I recommend saying tuberculosis, instead of TB, as it is the first (and only) time the disease is mentioned in the manuscript.

Checked and corrected

Line 503: Remove comma after 2730.

Checked and corrected

Line 534 and 536: Remove space between MapDamage and 2.

Checked and corrected

Line 559: Give the full form of GTDBtk.

Checked and corrected

Line 579: This line is missing a verb after "we" for e.g. downloaded or acquired.

Corrected to: "we downloaded..."

Lines 622-625: Please ensure that numbers that need to be shown as superscript are correctly formatted. For eg. $1.8 \cdot 10^{-6}$ needs to be corrected to 1.8×10^{-7} or given as superscript.

Checked and corrected